# Machines and Mathematical Mutations: Using GNNs to Characterize Quiver Mutation Classes

Jesse He [1 2]   Helen Jenne [2]   Herman Chau [3]   Davis Brown [2]   Mark Raugas [2]   Sara Billey [3]   Henry Kvinge [2 3]

## Abstract

Machine learning is becoming an increasingly valuable tool in mathematics, enabling one to identify subtle patterns across collections of examples so vast that they would be impossible for a single researcher to feasibly review and analyze. In this work, we use graph neural networks to investigate *quiver mutation*—an operation that transforms one quiver (or directed multigraph) into another—which is central to the theory of cluster algebras with deep connections to geometry, topology, and physics. In the study of cluster algebras, the question of *mutation equivalence* is of fundamental concern: given two quivers, can one efficiently determine if one quiver can be transformed into the other through a sequence of mutations? In this paper, we use graph neural networks and AI explainability techniques to independently discover mutation equivalence criteria for quivers of type $\tilde{D}$. Along the way, we also show that even without explicit training to do so, our model captures structure within its hidden representation that allows us to reconstruct known criteria from type $D$, adding to the growing evidence that modern machine learning models are capable of learning abstract and parsimonious rules from mathematical data.

## 1. Introduction

Examples play a fundamental role in the mathematical research workflow. Exploration of a large number of examples builds intuition, supports or disproves conjectures, and points the way towards patterns that are later formalized as theorems. While computer-aided simulation has long played

an important role in mathematics research, modern machine learning tools like deep neural networks have only recently begun to be more broadly applied. From the other direction, increasing attention in machine learning has been given to whether complex models like neural networks can learn to *reason* when faced with mathematical or algorithmic tasks. For example, the phenomenon of *algorithmic alignment* in graph neural networks is known to improve the sample complexity of such networks when compared to networks with similar expressive power (Dudzik & Veličković, 2022; Xu et al., 2020), and graph neural networks remain competitive with transformers for recognizing local subgraph structures (Sanford et al., 2024).

Of course, working mathematicians often need more than just a model that achieves high accuracy on a given task. In many cases, a model is only useful if it learns features that can provide insight to a mathematician. Further, one must be able to extract this insight from the model. In this work, we consider the specific problem of characterizing *quiver mutation equivalence classes*. We show first that in this setting a graph neural network learns representations that align with known non-trivial mathematical theory. We then show how one can use a performant model to generate a concise conjecture, which we then prove.

Introduced by Fomin and Zelevinsky in (Fomin & Zelevinsky, 2002), *quiver mutation* is a combinatorial operation on quivers (directed multigraphs) which arises from the notion of a cluster algebra, an algebraic construction with deep connections to geometry and physics. Quiver mutation defines an equivalence relation on quivers (that is, it partitions the set of quivers into disjoint subsets). Two quivers belong to the same equivalence class if we can apply some appropriate sequence of quiver mutations to the first and obtain the second. Identifying whether such a sequence of mutations exists is generally a hard problem (Soukup, 2023). In some cases, however, a concise characterization which involves checking some simple conditions is known. For example, Theorem 5.1 (Buan & Vatne, 2008) and Theorem 5.3 (Vatne, 2010) tell us that we can check whether a quiver belongs to types $A$ or $D$ by verifying certain conditions relating to the presence of specific structural motifs. Henrich (2011) provides a complete description of Dynkin and affine Dynkin type quivers in terms of certain families of infinite graphs.

[1]Halıcıoğlu Data Science Institute, University of California San Diego, San Diego, CA, USA [2]Pacific Northwest National Laboratory, Richland, WA, USA [3]Department of Mathematics, University of Washington, Seattle, WA, USA. Correspondence to: Jesse He <jeh020@ucsd.edu>, Henry Kvinge <henry.kvinge@pnnl.gov>.

*Proceedings of the $42^{nd}$ International Conference on Machine Learning*, Vancouver, Canada. PMLR 267, 2025. Copyright 2025 by the author(s).

We train a graph neural network (GNN) on a dataset consisting of $\sim 70,000$ quivers labeled with one of six different types ($A$, $D$, $E$, $\tilde{A}$, $\tilde{D}$, $\tilde{E}$). We find that not only does the resulting model achieve high accuracy, it also extracts features from type $D$ quivers that align with the characterization from (Vatne, 2010). We identify the latter through a careful application of model explainability tools and exploration of hidden activations. Pushing this further, we carefully probe the hidden representations of type $\tilde{D}$, independently achieving a similar characterization[1]. We describe how insights gained through the clustering of hidden activations and other explainability tools helped us prove our characterization of type $\tilde{D}$ quivers (Theorem 6.1). This provides yet another example of how machine learning can be a valuable tool for the research mathematician.

In summary, this work's contributions include the following: (1) We describe an application of graph neural networks to the problem of characterizing mutation equivalence classes of quivers. (2) Using AI explainability techniques, we provide strong evidence that our model learned features which align with human-developed characterizations of quivers of type $D$ from the mathematical literature. (3) Using insights gained from interpreting our model, we independently conjecture and then prove a characterization of quivers of type $\tilde{D}$ in terms of certain subquivers.

## 2. Background and Related Work

Quivers and quiver mutations are central in the combinatorial study of *cluster algebras*, a relatively new but active research area with connections to diverse areas of mathematics. For a high-level discussion of quiver mutation in the broader context of cluster algebras, see Appendix A.1. Since quivers and quiver mutation can be studied independently of their algebraic origin, we have written the rest of the paper so that it does not depend on this background.

### 2.1. Mutation-Finite Quivers

In (Fomin & Zelevinsky, 2003), Fomin and Zelevinsky gave a complete classification of finite cluster algebras, which can be generated by a finite number of variables. (Equivalently, their associated quiver mutation classes are finite). Amazingly, they correspond exactly to the Cartan-Killing classification of semisimple Lie algebras. Their result says that a quiver associated to a cluster algebra of finite type must be mutation equivalent to an orientation of a Dynkin diagram (Figure 3 for examples). However, this result does not give an algorithm for checking this. To answer this question, Seven (Seven, 2007) gave a full description of the associated

---

[1]After initially circulating our completed manuscript within the community, we were alerted that (Henrich, 2011) had already proved the theorem we discovered in somewhat different language.

quivers by computing all minimal quivers of infinite type. Since then, several other researchers have provided explicit characterizations of particular mutation classes of quivers (Bastian, 2011; Buan & Vatne, 2008; Vatne, 2010). Our main result follows these: we give an explicit characterization of quivers of type $\tilde{D}_n$, akin to the characterization of quivers of type $D_n$ given in (Vatne, 2010). This differs subtly from (Henrich, 2011), which characterizes finite-type quivers as subgraphs of certain families of infinite graphs.

### 2.2. Mathematics and Machine Learning

Machine learning has recently gained traction as a tool for mathematical research. Mathematicians have leveraged its ability to, among other things, identify patterns in large datasets. These emerging applications have included some within the field of cluster algebras (Cheung et al., 2023; Bao et al., 2020; Dechant et al., 2023). Unlike our work, this research does not aim to establish new theorems around mutation equivalence classes, focusing rather on the performance of models on different versions of this problem. Armstrong-Williams et al. (2025) obtain a result concerning the *mutation-acyclicity* of quivers, though their theoretical result guides their ML investigation rather than the reverse. Though unrelated to cluster algebras, Davies et al. (Davies et al., 2021) take an approach similar to the one taken here: using machine learning to guide mathematicians' intuition. They focus on two questions: one related to knot theory and one related to representation theory.

Due to the existence of unambiguous ground truth and known algorithmic solutions, there has also been renewed interest in using mathematical tasks to better analyze how machine learning models learn tasks at a mechanistic level, including the emergence of reasoning in large models. For example, in (Chughtai et al., 2023), the authors use group operations to investigate the question of *universality* in neural networks. Group multiplication is also used in (Stander et al., 2024) to investigate the *grokking* phenomenon. The idea of *mechanistic interpretability*—explaining model behavior by identifying the role of small collections of neurons—is also demonstrated in (Zhong et al., 2023), where Zhong et al. are able to recover two distinct algorithms from networks trained to perform modular arithmetic, and (Liu et al., 2023), where Liu et al. find evidence that a network trained to predict the product of two permutations learns group-theoretic structure.

## 3. Preliminaries

### 3.1. Quivers and Quiver Mutation

In their work on cluster algebras (Fomin & Zelevinsky, 2002), Fomin and Zelevinsky introduce the notion of *matrix mutation* on skew-symmetric (or skew-symmetrizable) in-

teger matrices. By regarding skew-symmetric matrices as directed graphs, we obtain a combinatorial interpretation of matrix mutation in terms of *quivers* which are the central objects of our study. In this section we briefly describe preliminaries concerning quivers and quiver mutations.

**Definition 3.1.** A *quiver* $Q$ is a directed (multi)graph with no loops or 2-cycles. There may be multiple parallel edges between vertices, represented as positive integer weights. The *underlying graph* of $Q$ is the undirected graph where we ignore edge orientations.

**Definition 3.2.** The *mutation* of a quiver $Q$ at a vertex $j$ is the quiver $\mu_j(Q)$ obtained by performing the following: (i) For each path $i \rightarrow j \rightarrow k$ in $Q$, add an arrow $i \rightarrow k$; (ii) Reverse all arrows incident to $j$; (iii) Remove any resulting 2-cycles created from the previous two steps.

Quiver mutation is an involution. That is, for a vertex $j$ in a quiver $Q$, $\mu_j(\mu_j(Q)) = Q$. Consequently, mutation is an equivalence relation on quivers.

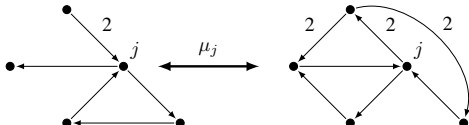

*Figure 1.* An example of a quiver and mutation at a vertex $j$.

**Definition 3.3.** We say two quivers $Q, Q'$ are *mutation equivalent* if $Q'$ can be obtained from $Q$ (up to isomorphism) by a sequence of mutations. We refer to the *mutation class* of $Q$ as the set $[Q]$ of (isomorphism classes of) quivers which are mutation-equivalent to $Q$.

**Definition 3.4.** We say a quiver $Q$ is *mutation-finite* if its mutation class $[Q]$ is finite, and *mutation-infinite* otherwise.

**Definition 3.5.** Given a starting quiver $Q$, the *mutation depth* of a quiver $Q' \in [Q]$ (with respect to $Q$) is the minimum number of mutations required to obtain $Q'$ from $Q$.

We will consider the mutation classes of quivers that are of *simply laced* (that is, with no parallel edges) Dynkin or extended (affine) Dynkin type. These are the quivers whose underlying undirected graphs are shown in Figure 3. In particular, the quivers of type $D$ and type $\tilde{D}$ will be the focus of our explainability analysis. We also include quivers of type $E$ (Figure 2) which are not finite or affine type in our training and test sets. (The diagram $E_n$ is only finite type for $n = 6, 7, 8$, and affine for $n = 9$.)

The following well-known lemma (Vatne, 2010) ensures the mutation classes of the simply laced Dynkin diagrams are well-defined.

**Lemma 3.6.** *If quivers $Q_1$ and $Q_2$ have the same underlying graph $T$ and $T$ is a tree, then $Q_1$ and $Q_2$ are mutation equivalent.*

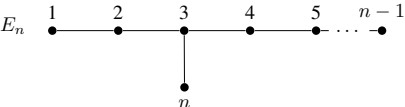

*Figure 2.* Coxeter-Dynkin diagram for $E_n$, $n \geq 6$. Quivers of type $E_n$ are only mutation-finite for $n = 6, 7, 8, 9$.

**The machine learning task:** Train a classifier $\Phi$ to predict the mutation class of a quiver of type $A$, $D$, $E$, $\tilde{A}$, $\tilde{D}$, or $\tilde{E}$ (Figure 3). We train $\Phi$ on a set of quivers with 6, 7, 8, 9, or 10 nodes, and test on quivers with 11 nodes. More details are provided in Section 4.1.

As we will show through our explainability studies, a model trained on this task can learn rich features that point towards concise characterizations of these quivers.

### 3.2. Graph Neural Networks

Because quivers are represented as directed graphs, it is natural to use graph neural networks to classify them. Graph neural networks (GNNs), introduced in (Defferrard et al., 2016; Kipf & Welling, 2017), are a class of neural networks which operate on graph-structured data via a *message-passing* scheme. Given an (attributed) graph $G = (V, E)$ with node features $x_v \in \mathbb{R}^p$ for each node $v \in V$ and $e_{uv} \in \mathbb{R}^q$ for each edge $(u, v) \in E$, each layer of the network updates the node feature by aggregating the features of its neighbors. The final graph representation is computed by pooling the node representations. Because prior work has characterized quiver mutation classes based on the presence of particular subgraphs, we use the most expressive GNN architecture for recognizing subgraphs (Xu et al., 2019). To this end, we adopt a version of the graph isomorphism network (GIN) introduced in (Xu et al., 2019) and modified in (Hu et al., 2020) to support edge features. Since quivers are directed graphs, we adopt a directed message-passing scheme with separate message-passing functions along each orientation of an edge. We refer to our architecture as a **Dir**ected **G**raph **I**somorphism **N**etwork with **E**dge features (DirGINE), and denote the network itself by $\Phi$. We describe our DirGINE architecture in greater detail in Appendix B.1.

## 4. Methods

### 4.1. Model Training

We train a 4-layer DirGINE GNN with a hidden layer width of 32 to classify quivers into types $A$, $D$, $E$, $\tilde{A}$, $\tilde{D}$, and $\tilde{E}$. The training data consists of quivers of each type on 6, 7, 8, 9, and 10 nodes. The test set consists of quivers of types $A, D, E, \tilde{A}, \tilde{D}$ on 11 nodes. (Type $\tilde{E}$ is not defined on 11 nodes.) We generate data with Sage (The Sage Developers, 2023; Musiker & Stump, 2011), which we de-

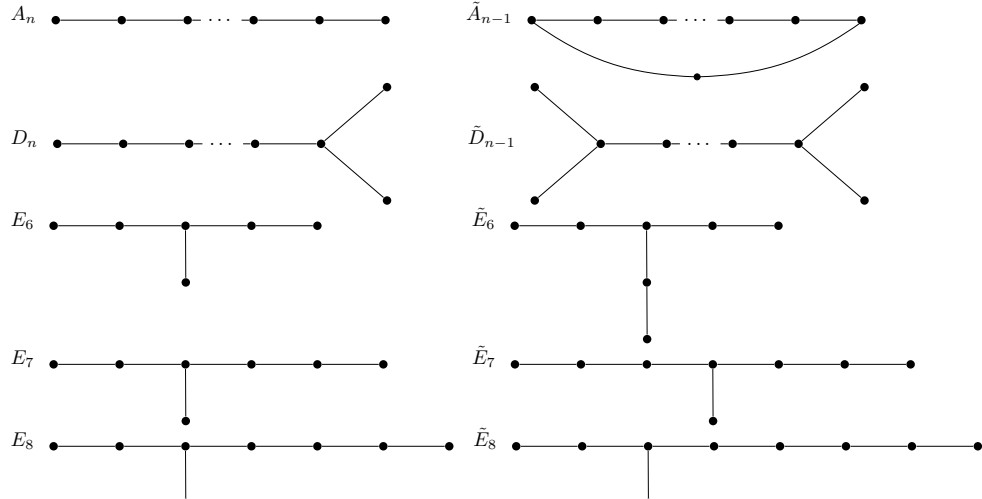

*Figure 3.* Simply laced Dynkin diagrams and their extensions.

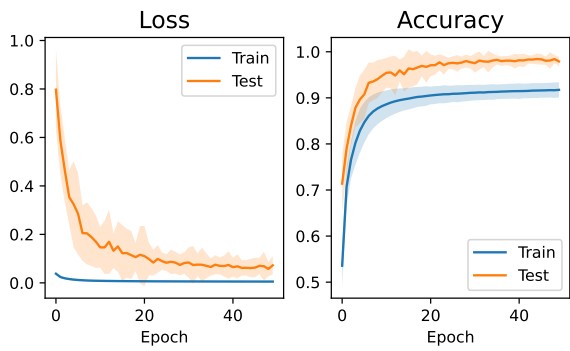

*Figure 4.* Average cross-entropy loss (left) and classification accuracy (right) on train and test sets across 10 trials of training. Testing accuracy is consistently higher than training accuracy, perhaps due to the absence of class $\tilde{E}$ in the test set and the fact that $\tilde{E}_8 = E_9$ in the train set.

scribe in greater detail in Appendix C. Figure 4 shows the average cross-entropy loss and classification accuracy by epoch across 10 trials. We take the best epoch from training (99.2% test accuracy) for our analysis.

While the differences between the train and test set (particularly the absence of type $\tilde{E}$ from the test set) might be problematic if our goal was to assess whether a machine learning model can classify quivers into mutation types, our primary goal is to extract mathematical insights from the features the model learns for types $D$ and $\tilde{D}$. As such, we use the test set to indicate whether a model was sufficiently performant to justify the application of explainability tools. Moreover, the inclusion of types $E$ and $\tilde{E}$ help modulate the difficulty of the classification problem, ensuring that the model learns discriminative features for classes $D$ and $\tilde{D}$.

### 4.2. Size Generalization

By training on quivers of sizes 7-10 and testing on quivers of size 11, we encourage our DirGINE to learn size-generalizable features, leveraging the promise of size generalization in GNNs. Given this, it is natural to ask how well our model performs for larger $n$. (After all, any classification rules for quivers should be size-generalizable as well.) To examine this question, we test our model on quivers of types $A$, $D$, $E$ $\tilde{A}$, and $\tilde{D}$ on $n = 12, 13, \ldots, 20$ vertices. Because the number of distinct quivers grows quickly with size, we only use a subsample of each class, which we discuss in greater detail in Appendix C. The results (Table 1) indicate that our GNN generalizes well, albeit not perfectly. We believe this is because our GNN has a fixed depth of 4, and hence cannot recognize the larger substructures that may appear in larger quivers. In particular, we will see in Section 5.2 that some quivers can be distinguished by long cycles, which message-passing networks may fail to recognize (Chen et al., 2020).

### 4.3. Explaining GNNs

In order to extract mathematical insight from a trained GNN model $\Phi$, we require a way to *explain* its predictions by identifying the substructures that are responsible for its predictions. That is, for each graph $G$, we wish to identify a small subgraph $G_S$ such that $\Phi(G) \approx \Phi(G_S)$. We use the GNN explanation method PGExplainer (Luo et al., 2020), which trains a neural network $g$ to identify important subgraphs. For an input graph $G$, the explanation network produces an attribution $\omega_{u,v}$ for each edge $(u, v)$ using the final representations for nodes $u$ and $v$.

While PGExplainer's effectiveness is mixed across different comparisons (Agarwal et al., 2023; Amara et al., 2022),

| $n$ | 12 | 13 | 14 | 15 | 16 | 17 | 18 | 19 | 20 |
|---|---|---|---|---|---|---|---|---|---|
| Accuracy | 99.6 | 98.7 | 97.7 | 95.5 | 94.3 | 92.0 | 91.1 | 89.4 | 89.1 |

*Table 1.* Accuracy of our trained DirGINE on unseen quivers of size $n = 12, 13, \ldots, 20$.

it is effective at providing model-level substructure explanations for graph classification tasks. For example, when applied to a GNN trained on the MUTAG dataset (Debnath et al., 1991) to predict the mutagenicity of molecules, PGExplainer is regularly able to identify that the model predicts mutagenicity based on the presence of nitro ($NO_2$) groups (Luo et al., 2020). As we will see in Section 5, the ability of PGExplainer to identify explanatory graph motifs makes it suitable for our purposes. To analyze our trained DirGINE, we train PGExplainer on 1000 randomly selected instances from the train set for 5 epochs. Further discussion of PGExplainer is provided in Appendix B.2.

# 5. Extracting Quiver Characterizations

Before we state Theorem 6.1, which we re-discover through analysis of our GNN model, we describe the known characterizations of mutation class $A$ quivers (Buan & Vatne, 2008) in Section 5.1 and mutation class $D$ quivers (Vatne, 2010) in Section 5.2. We also describe in Section 5.2 how we can reconstruct the characterization of type $D$ (Vatne, 2010) by probing our trained GNN[2].

## 5.1. The Mutation Class of $A_n$ Quivers

The class of $A_n$ quivers consists of all quivers which are mutation equivalent to (1).

$$\underset{1}{\bullet} \longrightarrow \underset{2}{\bullet} \longrightarrow \cdots \longrightarrow \underset{n-1}{\bullet} \quad \underset{n}{\bullet} \qquad (1)$$

The following theorem provides a combinatorial characterization of all such quivers. We refer to the collection of all such quivers as $\mathcal{M}_n^A$. We will also denote $\mathcal{M}^A = \bigcup_{n \geq 1} \mathcal{M}_n^A$.

**Theorem 5.1** (Buan & Vatne 2008). *A quiver $Q$ is in the mutation class $\mathcal{M}_n^A$ if and only if: (i) All cycles are oriented 3-cycles. (ii) Every vertex has degree at most four. (iii) If a vertex has degree four, two of its edges belong to the same 3-cycle, and the other two belong to a different 3-cycle. (iv) If a vertex has degree three, two of its edges belong to a 3-cycle, and the third edge does not belong to any 3-cycle.*

While this result is interesting in its own right, the importance for this paper is that the quivers in the mutation class of type $D_n$ contain subquivers of type $A$.

_______________

[2]Unlike Theorem 6.1, we were already aware of the type $A$ and type $D$ characterizations when we began this work.

## 5.2. The Mutation Class of $D_n$ Quivers

We now describe the classification of the mutation class $\mathcal{M}_n^D$ of Type $D_n$ quivers by Vatne (2010). As with type $A$, we use $\mathcal{M}^D$ to denote the collection of type $D$ quivers for all $n \geq 3$ (noting that $A_3 = D_3$ and that $D_1, D_2$ are not defined):

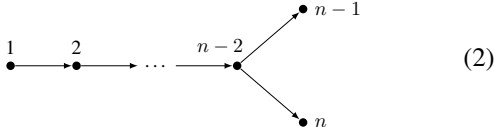

$$(2)$$

In Vatne's classification, each type is a collection of subquivers joined by gluing vertices. The proof relies on the fact that if the vertex joining the blocks is a *connecting vertex* (defined below), one can mutate a sub-quiver of type $A$ to the quiver in (1) without ever mutating at the connecting vertex. Formally, we have the following.

**Definition 5.2.** For a quiver $\Gamma \in \mathcal{M}_n^A$, we say a vertex $c$ of $\Gamma$ is a *connecting vertex* if $c$ is either degree one or degree two and part of an oriented 3-cycle.

Vatne proves the following classification of quivers of $\mathcal{M}_n^D$ into four types by first proving that each subtype is mutation-equivalent to (2), then proving that the collection of quivers described by these subtypes is closed under quiver mutation.

**Theorem 5.3** (Vatne 2010). *The quivers of the mutation class of $D_n$ consist of four subtypes shown in the diagrams below. In these diagrams, $\Gamma$, $\Gamma'$, and $\Gamma''$ are full subquivers of mutation class $A$ with a connecting vertex. Unoriented edges mean that the orientation does not matter.*

*Type I.*

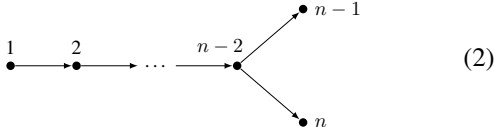

$$(3)$$

*Type II.*

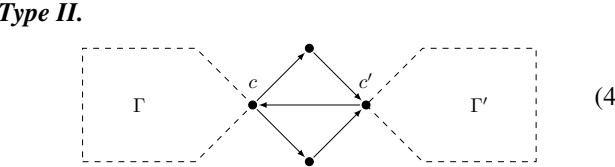

$$(4)$$

*Type III.*

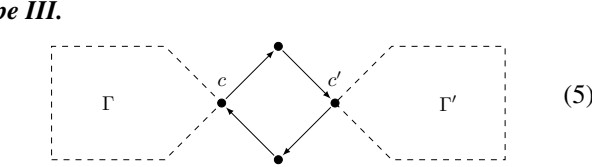

$$(5)$$

*Type IV.*

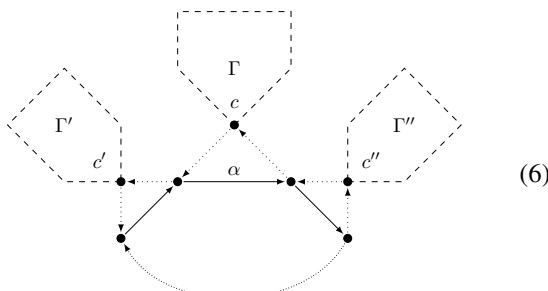

$$\text{(6)}$$

In particular, we draw attention to the oriented *central cycle* in type IV. Every edge in the central cycle may be part of an oriented triangle with a connecting vertex not on the central cycle (called a *spike*).

Because our DirGINE can, in general, learn to recognize certain subgraphs in a quiver, it is reasonable to ask whether the model was relying on the same subtype motifs identified by human mathematicians. We investigated the case of the type $D$ classification from Theorem 5.3 using PGExplainer as described in Section 4.3. Some sample soft masks produced by PGExplainer are shown for quivers correctly classified as type $D$ in Figure 5. Dark red edges are judged more important for the type $D$ prediction by PGExplainer, while lighter edges are judged less important. In each case, the explanation highlights edges which align with Theorem 5.3. Figure 5 contains an example of each of the four types. The reader can check that in each case the edges of the relevant subtype motif tends to have substantially higher attribution. This initial analysis seems to suggest that our model has independently learned the same subtype motifs for classifying type $D$ quivers from known (human) theory. The attributions shown in Figure 5 also suggest that the model relies on the opposite end of the quiver—a behavior that seems superfluous for recognizing type $D$. We will show in Section 6 that this is actually very important for distinguishing quivers of type $D$ from $\tilde{D}$.

Figure 5 strongly suggests that our GNN recognizes the same subtypes as in Theorem 5.3. However, one should be careful in this interpretation, as there is a substantial literature showing that it is easy to misinterpret post-hoc explainability methods (Ghorbani et al., 2019; Kindermans et al., 2019). Thus, we also examine the embeddings of type $D_n$ quivers in the model's latent space. We use principal component analysis (PCA) to reduce the dimension of the embedding from the model width of 32 to 2 dimensions for visualization. The resulting graph embeddings, plotted in Figure 6, show a clear separation of the different subtypes. In fact, the layer 3 embeddings in the original 32-dimensional embedding space can be separated by a linear classifier with $99.7 \pm 0.0\%$ accuracy. Subtypes I through IV are not labeled in the training data, so this analy-

sis, combined with the PGExplainer attributions, provides strong evidence that a GNN is capable of re-discovering the same abstract, general characterization rules that align with known theory through training on a naive classification task.

### 5.2.1. DO CHANGES IN SUBTYPE MOTIFS ACTUALLY IMPACT PREDICTIONS?

To further understand how the model is using the type $D$-specific subquivers from Theorem 5.3 in practice, we examine the model's predictions when the edges identified by PGExplainer are removed. If the model is primarily keying into the type $D$ motif, removing this should result in the quiver being predicted as type $A$.

We find that across all $32,066$ test examples from type $D$, a plurality (14,916 or 46.5%) of the predictions flip to $A$, as we would expect if it was using the characterization from Theorem 5.3. Of the remaining examples, most (14,238 or 44.4% of the total) flip to a predicted class of $E$, with the next-largest being $D$ (no flip) at 2,581 or 8.0%. Finally, 264 quivers (0.08%) flip to $\tilde{A}$, while 67 quivers (0.02%) flip to $\tilde{D}$. None of the predictions flip to $\tilde{E}$.

Why are there so many instances that flip to type $E$? This may be due to the PGExplainer attributions for type $D$ being inexact, perhaps because PGExplainer generalizes imperfectly or because some edges do not contribute positively to $D$ but rather contribute negatively to other classes. As a result, many of the quivers where we remove highly attributed edges may be out-of-distribution for the model. Since type $E$ is the only class which contains mutation-infinite quivers, it is perhaps not surprising that the model would predict these out-of-distribution quivers are of type $E$.

## 6. Characterizing $\tilde{D}$ Quivers

In this section, we describe how our trained model and explainability techniques enable us to independently discover a characterization of the mutation class of $\tilde{D}$ quivers, stated in Theorem 6.1 below. We learned after initial circulation of this work that this result can be derived from (Henrich, 2011). However, as our characterization closely reflects the way in which we analyzed our model, we felt it would still be of interest to the machine learning community as an example of extracting research-level mathematics from a deep learning model. Due to the complexity of the characterization, some of the details of the characterization are left to Appendix D, and the proof to Appendix E.

**Theorem 6.1.** *The mutation class of class $\tilde{D}_{n-1}$ quivers is $\mathcal{M}_{n-1}^{\tilde{D}}$, the collection of quivers of paired types together with Types V, Va, Vb, V', Va', Vb', VI, and VI'.*

Similar to Vatne's classification of the mutation class of $D_n$ quivers, our classification consists of different subtypes.

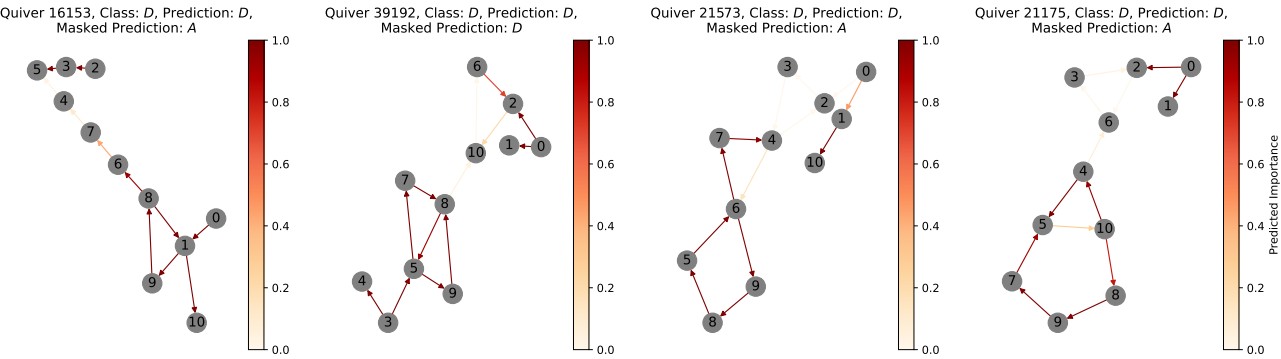

*Figure 5.* Edge attributions from PGExplainer on type $D_{11}$ quivers of each subtype. The masked prediction is the GNN prediction when highly attributed edges are removed. From left to right: **Type I.** Five dark red edges highlight the subquiver consisting of the leaves 0 and 10, the connecting vertex 1, and the 3-cycle that the connecting vertex is a part of; **Type II.** Five dark red edges highlight the center block seen in the Type II diagram(the vertices 5 and 8 are $c$ and $c'$, respectively); **Type III.** Four dark red edges highlight the oriented 4-cycle, where the vertex 6 is the vertex $c'$; **Type IV.** The dark red edges highlight the oriented 5-cycle and the spike $5 \xrightarrow{\alpha} 10 \to 4 \to 5$.

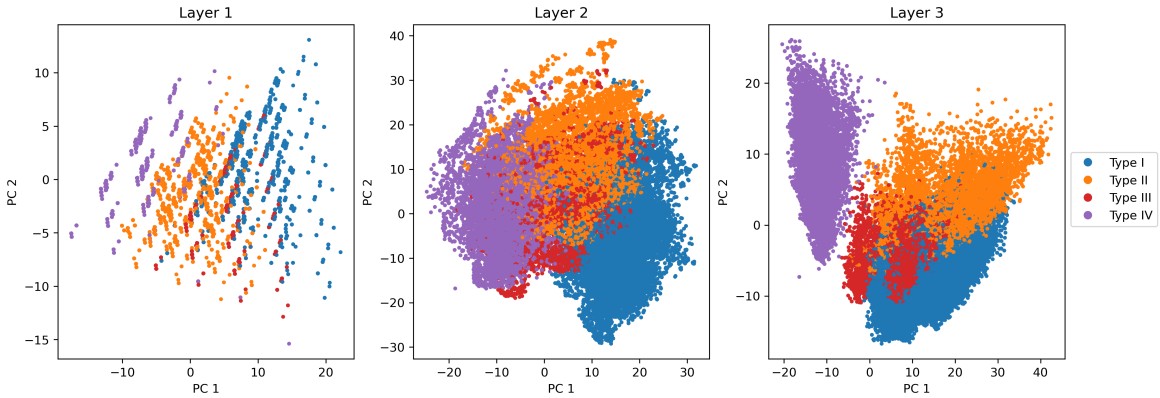

*Figure 6.* PCA of latent space embeddings for mutation class $D$ quivers colored by subtypes.

However, there are many more subtypes compared to the type $D$ case, so we find it convenient to organize them into families: what we call *paired types*, quivers with one central cycle, and quivers with two central cycles.

As in Types $A$ and $D$, Lemma 3.6 means that we may choose an arbitrary orientation of the extended Dynkin diagram $\tilde{D}_{n-1}$. It will be convenient to begin with the orientation in (7), viewing it as two quivers $Q_1$ and $Q_2$ of type $D$ (2) connected at their roots by a connecting vertex $c$.

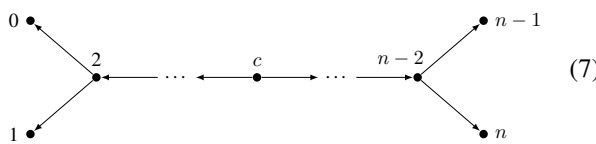

$$\tag{7}$$

From the orientation in (7) it is immediately clear that by mutating $Q_1$ and $Q_2$ independently without mutating $c$, we

can obtain any pair of subtypes of type $D$. Because the placement of $c$ is arbitrary, we see that many type $\tilde{D}_{n-1}$ quivers can be described by two of the type $D$ subtypes characterized in Section 5.2 which share a type $A$ piece $\Gamma_c$. We will refer to such quivers as Types I-I, I-II, I-III, etc., and collectively as *paired types*. (See Figure 13 in Appendix F for all paired types.) It remains, then, to identify the quivers in this mutation class which not are of paired type.

While a human mathematician could conceivably discover the same characterization of $\tilde{D}$ quivers simply by beginning with (7) and exhaustively performing mutations, the mutation class of $\tilde{D}$ quivers admits many diverse subtypes compared to classes $A$ or $D$. This increased complexity creates some difficulty (and perhaps more importantly, tedium) in examining examples manually. By taking advantage of machine learning, we are able to quickly organize examples into distinct families to examine.

Based on our strategy in Section 5.2, we plot PCA reduc-

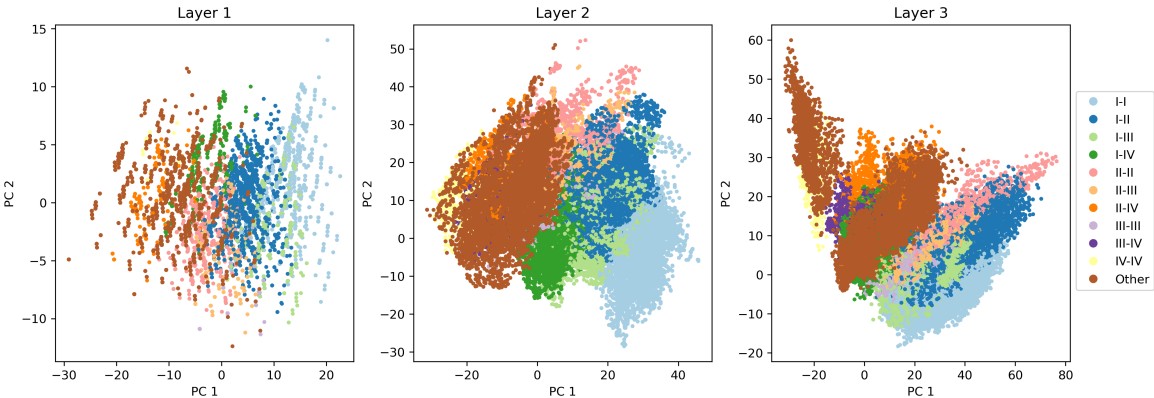

*Figure 7.* PCA reductions of latent space embeddings for mutation class $\tilde{D}_{10}$ quivers colored by paired types or "Other".

tions of the latent space in Figure 7. We can see that the quivers that do not correspond to paired subtypes, colored as "Other", separate clearly into two clusters in layer 3. By isolating these quivers and performing $k$-means clustering with $k = 2$, the model guides our characterization of the remaining class $\tilde{D}$ subtypes. Figure 8 shows examples from each cluster. (More examples are given in Appendix F.) The key insight we gain from examining the quivers in each cluster is that the remaining subtypes can be separated by the number of Type IV-like central cycles.

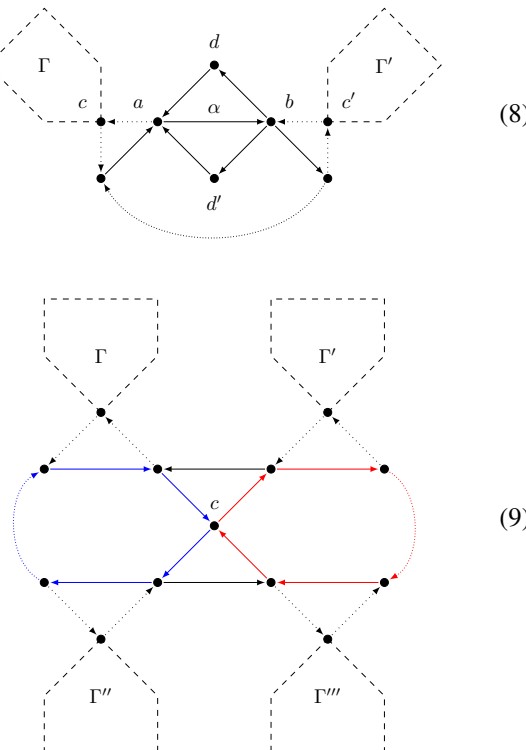

$$(8)$$

The type $\tilde{D}$ quivers with a single central cycle make up the *Type V family*. These all have a sort of "double spike" motif;

$$(9)$$

we show one example of *Type V* in (8). The rest of the Type V family, types Va, Vb, V', Va', and Vb', can be obtained from (8), as we describe in Appendix D.1.

The *Type VI family* consists of the type $\tilde{D}$ quivers with two central cycles. An example of *Type VI* is provided in (9), where we color each central cycle for clarity. Type VI' (10) is an exceptional version of type VI in which both central cycles are of length 3 and $c$ is allowed to be a connecting vertex for a type $A$ quiver. Theorem 6.1, which we prove in Appendix E, states that these subtypes together with the paired types give an exhaustive characterization of the mutation class of $\tilde{D}_{n-1}$ quivers.

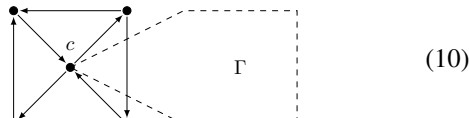

$$(10)$$

## 7. Conclusion

In this work we analyze a graph neural network trained to classify quivers as belonging to one of 6 different types, motivated by the theory of cluster algebras and the problem of quiver mutation equivalence. Using explainability techniques, we provide evidence that the model learns prediction rules that align with existing theory for one of these types (type $D$). Moreover, the model behavior which allows us to recover this result emerges from the model in an unsupervised manner—the model is not given any subtype labels, and yet is able to identify relevant blocks to recognize type $D$ quivers. Applying the same explainability techniques to another case, we also independently discover and prove a characterization of the mutation class of $\tilde{D}_{n-1}$ quivers. Taken together, our work provides more evidence supporting the idea that machine learning can be a valuable tool in the mathematician's workflow by identifying novel patterns in mathematical data.

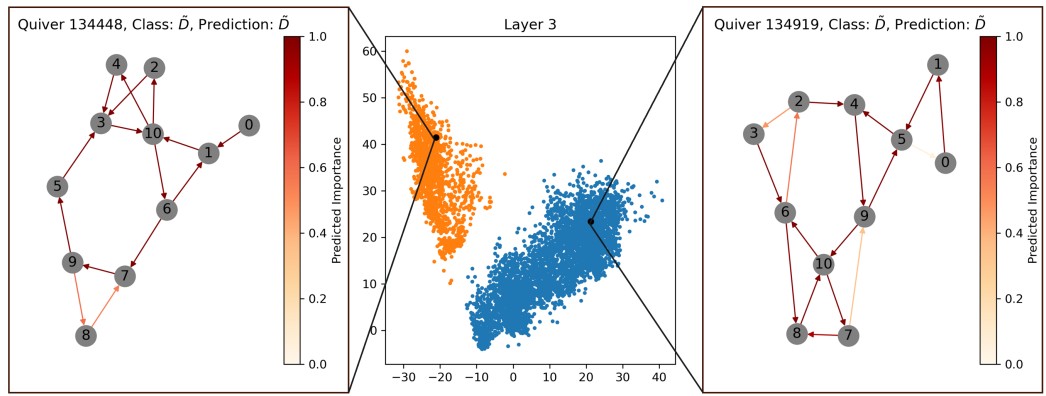

*Figure 8.* PCA of clustered layer 3 latent space embeddings of "Other" quivers in $\mathcal{M}_{10}^{\tilde{D}}$ (middle), with selected examples from each cluster (left, right). Edges are colored by PGExplainer attributions. The quiver on the left is of Type V while the quiver on the right is of Type VI.

## Impact Statement

This work showcases an application of explainability to advance an application in algebraic combinatorics. It shares the societal consequences of machine learning and algebraic combinatorics in general, none which we feel must be specifically highlighted here.

## Acknowledgements

The authors would like to thank Scott Neville and Kayla Wright for helpful discussions. We also thank Pavel Tumarkin for bringing to our attention the work of Henrich (2011). This research was supported by the Mathematics for Artificial Reasoning in Science (MARS) initiative via the Laboratory Directed Research and Development (LDRD) investments at Pacific Northwest National Laboratory (PNNL). PNNL is a multi-program national laboratory operated for the U.S. Department of Energy (DOE) by Battelle Memorial Institute under Contract No. DE-AC05-76RL0-1830.

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

# A. Additional Background

## A.1. Cluster algebras and quiver mutations

A cluster algebra is a special type of commutative ring that is generated (in the algebraic sense) via a (possibly infinite) set of generators that are grouped into *clusters*. A cluster algebra may have finitely or infinitely many generators, but the size of each cluster is always finite and fixed. A cluster algebra is said to be of *rank* $n$ if each of the clusters contains $n$ generators, called *cluster variables*. These clusters are related via an *exchange property* which tells us how to transform one cluster to another (Fomin & Zelevinsky, 2002).[3] It turns out that there is a nice combinatorial interpretation of this transformation when we interpret clusters as quivers with each generator corresponding to a vertex in the quiver. Then quiver mutation describes this exchange of cluster variables. In this setting, the mutation equivalence problem asks when two clusters generate the same cluster algebra.

Quiver mutation also appears in physics in the form of *Sieberg duality*. Seiberg duality is an important low-energy identification of naively distinct non-Abelian four dimensional $N = 1$ supersymmetric gauge theories, where gluons and quarks of one theory are mapped to non-Abelian magnetic monopoles of the other, and vice versa, but result non-trivially in the same long-distance (low-energy) physics (Seiberg, 1995). Quiver gauge theories in string theory are $N = 1$ supersymmetric quantum field theories that have field and matter field content defined in terms of their superpotential read off from an associated quiver diagram. They arise in a number of scenarios, including the low energy effective field theory associated to D-branes at a singularity. The superpotential and its associated quiver are defined by the representation theory of the finite group associated to the singularity (e.g., if it is of ADE type), via the McKay correspondence (Greene et al., 1999). Associated to quivers are cluster algebras, and quiver mutations in this physical context are maps that identify naively distinct $N = 1$ $4d$ supersymmetric gauge theories under Sieberg duality in the low-energy (IR) limit. There is an extensive literature in this area of physics, some earlier work has used machine learning techniques to study pairs of candidate Seiberg dual physical theories related by quiver mutations (Bao et al., 2020). We hope that the present work may be helpful for those working to develop an improved mathematical understanding of both Seiberg duality and quiver gauge theory in general.

## A.2. Mutation-Finite Quivers

Quivers of type $\tilde{D}_n$—the main subject of our study—are *mutation-finite*. That is, they have a finite mutation equivalence class. Mutation-finite quivers and their associated cluster algebras are of interest to many cluster algebraists. Felikson, Shapiro, and Tumarkin 2012b gave a description of the mutation-finite quivers in terms of *geometric type* (those arising from triangulations of bordered surfaces), the $E_6, E_7, E_8$ Dynkin diagrams and their extensions, and two additional exceptional types $X_6$ and $X_7$ identified by Derksen and Owen (Derksen & Owen, 2008). Specifically, they showed that mutation-finite quivers must either be decomposable into certain *blocks* or contain a subquiver which is mutation equivalent to $E_6$ or $X_6$. It will be occasionally useful to refer to these blocks in in Appendix D, so we describe them here. However, block decompositions are not unique in general, so we will not place too much emphasis on the block decompositions of each quiver we describe.

**Definition A.1** (Felikson et al. 2012a). A *block* is one of six graphs shown in Figure 9, where each vertex is either an *outlet* or a *dead end* (Fomin et al., 2008). A connected quiver $Q$ is *block-decomposable* if it can be obtained by gluing together blocks at their outlets, such that each vertex is part of at most two blocks. Formally, one constructs a block-decomposable quiver as follows:

  (i) Take a partial matching of the combined set of outlets (no outlet may be matched to an outlet from the same block);

  (ii) Identify the outlets in each pair of the matching;

  (iii) If the resulting quiver contains a pair of edges which form a 2-cycle, remove them.

It is worth noting that classification in the mutation-finite setting has proven to be more challenging than in the finite setting. The classification of *mutation-finite* cluster algebras in the case with no frozen variables was achieved nearly a decade after Fomin and Zelevinsky classified finite cluster algebras (Felikson et al., 2012b;a), and the general case was solved only last year (Felikson & Tumarkin, 2024).

---

[3]In general, a cluster consists of both cluster variables and generators known as frozen variables (that lack this exchange property).

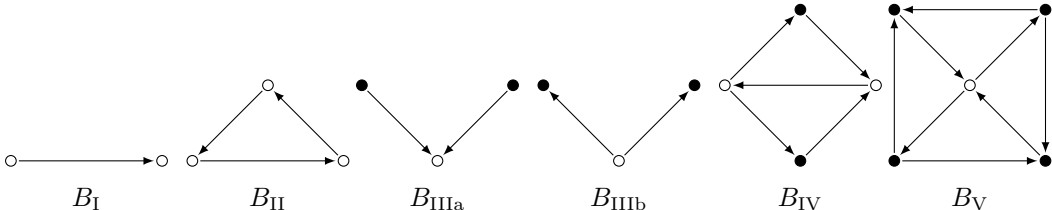

*Figure 9.* Blocks of type I-V introduced by (Fomin et al., 2008). Open circles denote *outlets*, which may be identified with at most one outlet from another block. Closed circles represent *dead ends*, which may not be identified with any other vertex.

## B. Implementation Details

### B.1. Model Architecture

Here we describe in detail our Directed Graph Isomorphism Network with Edge Features (DirGINE). The DirGINE uses a message-passing scheme, maintaining a representation of each node. In each layer, each node's representation is updated according to a parameterized function of its neighbors' representations. Formally, the $\ell$-th layer is given by

$$x_v^{(\ell)} = \text{ReLU}\left(W^{(\ell)}x_v^{(\ell-1)} + \sum_{(u,v)\in E} \varphi_{\text{in}}^{(\ell)}\left(x_u^{(\ell-1)}, e_{uv}\right) + \sum_{(v,w)\in E} \varphi_{\text{out}}^{(\ell)}\left(x_w^{(\ell-1)}, e_{vw}\right)\right) \tag{11}$$

where $W^{(\ell)}$ is an affine transformation and $\varphi_{\text{in}}^{(\ell)}$ and $\varphi_{\text{out}}^{(\ell)}$ are feedforward neural networks with 2 fully connected layers. Because we are classifying graphs, we use *sum pooling*. That is, in the final layer $L$ we can assign a vector to the entire graph $G$ by adding the vectors associated with each vertex in the layer. We write

$$\Phi(G) = \Phi^{(L)}(G) = \sum_{v\in V(G)} x_v^{(L)}. \tag{12}$$

The expressive power of graph neural networks is intimately connected to the classical Weisfeiler-Lehman (WL) graph isomorphism test (Weisfeiler & Leman, 1968). Given an undirected graph with constant node features and no edge features, a graph neural network cannot distinguish two graphs which are indistinguishable by the WL test (Xu et al., 2019), and graph neural networks are able to count some (but not all) substructures (Chen et al., 2020). In our case, operating on directed graphs with edge features slightly enhances the expressive power of our network, as the WL tests for attributed and directed graphs is strictly stronger than the undirected WL test (Beddar-Wiesing et al., 2022). As we saw in Section 5, the ability to distinguish directed substructures is crucial to their application in classifying quiver mutation classes. (For example, distinguishing a Type $D_n$ from a Type $\widetilde{A}_{n-1}$ quiver.)

### B.2. Further Discussion of PGExplainer

While a number of post-hoc explanation methods exist for GNNs, most fall into one of two categories:

(i) *Gradient-based* methods use the partial derivatives of the model output with respect to input features. A larger gradient is assumed to mean that a feature is more important.

(ii) *Perturbation-based* methods observe how the model's predictions change when features are removed or distorted. Larger changes indicate greater importance.

The method we adopt, PGExplainer (Luo et al., 2020), is a perturbation-based method which produces edge attributions using a neural network $g$. Using the final node embeddings of $u$ and $v$ as well as any edge features $e_{uv}$, $g$ produces an attribution

$$\omega_{uv} = g(x_u^{(L)}, x_v^{(L)}, e_{uv}). \tag{13}$$

We use the implementation provided by PyTorch Geometric (Fey & Lenssen, 2019), where $g$ is implemented as an MLP followed by a sigmoid function to ensure that $0 \leq \omega_{uv} \leq 1$. Then rather than produce a "hard" subgraph as our explanatory

graph $G_S$, the attribution matrix $\Omega = (\omega_{ij})$ can be seen as a "soft" mask for the adjacency matrix $A(G)$. That is, instead of providing a binary 0-1 attribution for each edge, PGExplainer provides an attribution $\omega_{ij} \in [0, 1]$. We then use the weighted graph with adjacency matrix $\Omega \odot A(G)$ for $G_S$ (where $\Omega \odot A(G)$ is the elementwise product).

PGExplainer follows prior work (Ying et al., 2019) in interpreting $G_S$ as a random variable with expectation $\Omega = \mathbb{E}[A(G_S)]$, where each edge $(i, j)$ is assigned a Bernoulli random variable with expectation $\omega_{ij}$. PGExplainer then attempts to maximize the *mutual information* $I(\Phi(G), G_S)$. However, because this is intractable in practice, the actual optimization objective is

$$\min_{\Omega} \mathrm{CE}(\Phi(G), \Phi(G_S)) + \alpha \|\Omega\|_1 + \beta H(\Omega). \tag{14}$$

Here $\mathrm{CE}(\Phi(G), \Phi(G_S))$ is the cross-entropy loss between the predictions $\Phi(G)$ and $\Phi(G_S)$, $\|\Omega\|_1$ is the $L_1$-norm of $\Omega$,

$$H(\Omega) = - \sum_{(i,j) \in E} \sum_{(i,k) \in E} [(1 - \omega_{ij}) \log(1 - \omega_{ik}) + \omega_{ij} \log(\omega_{ik})], \tag{15}$$

and $\alpha$ and $\beta$ are hyperparameters. In our analysis, we use hyperparameter values $\alpha = 2.5$ and $\beta = 0.1$. The $\|\Omega\|_1$ term acts as a size constraint, penalizing the size of the selected $G_S$. The $H(\Omega)$ term acts as a connectivity constraint, penalizing instances where two incident edges are given very different attributions. By training a neural network to compute $\Omega$, PGExplainer allows us to generate explanations for new graphs very quickly, as well as take a more global view of the model behavior.

## C. Data Generation and Model Training

Quivers were generated using Sage (The Sage Developers, 2023; Musiker & Stump, 2011). For training and inference, each quiver was converted to PyTorch Geometric (Fey & Lenssen, 2019). Following the representation convention in Sage, $k$ parallel edges are represented by a single edge with edge attribute $(k, -k)$, and each vertex is initialized with constant node feature. The train set consists of:

- All quivers of types $A$, $D$, $\tilde{A}$, and $\tilde{D}$ on 7, 8, 9, and 10 nodes.

- All quivers of type $\tilde{E}$. (Type $\tilde{E}$ is only defined for 7, 8, and 9 nodes, corresponding to extended versions of $E_6$, $E_7$, and $E_8$, respectively. All quivers of type $\tilde{E}$ are mutation-finite.)

- All quivers of type $E$ for $n = 6, 7, 8$. (The Dynkin diagram $E_9$ is the same as the extended diagram $\tilde{E}_8$.) Type $E$ is only mutation-finite for $n = 6, 7, 8$. and coincides with $\tilde{E}_8$ for $n = 9$.

- Quivers of type $E_{10}$ up to a mutation depth of 8, with respect to Sage's standard orientation for $E_{10}$ (Figure 10). (While type $E$ is mutation finite for $n \leq 9$, $E_{10}$ is mutation-infinite).

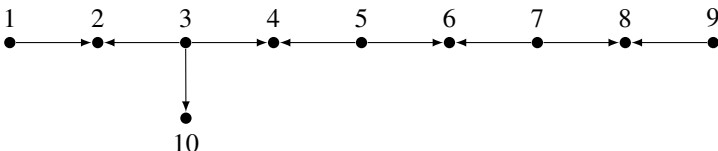

*Figure 10.* Default orientation of $E_{10}$ in Sage. Mutation depth is assessed with respect to this orientation for generating data in Sage.

The test set consists of quivers on 11 nodes. We use all quivers of type $A_{11}, \tilde{A}_{10}, D_{11}$ and $\tilde{D}_{10}$, and again generate quivers up to a mutation depth of 8 for $E_{11}$. The number of quivers of each size from each class can be found in Table 2. Note that type $\tilde{E}$ is absent from the test set, because $\tilde{E}$ is not defined for 11 nodes. The class of type $\tilde{A}$ quivers is also unique, as the collection of $\tilde{A}_{n-1}$ is actually partitioned into $\lfloor n/2 \rfloor$ distinct mutation classes, as described by (Bastian, 2011).

For the size generalization experiment in Section 4.2, we only generate quivers up to a mutation depth of 6, as the number of distinct quivers grows too quickly with size to generate classes exhaustively. When the number of generated quivers exceeds 100,000, we randomply subsample 100,000 quivers to avoid out-of-memory errors.

We train with the Adam optimizer for 50 epochs with a batch size of 32 using cross-entropy loss with $L_1$ regularization ($\gamma = 5 \times 10^{-6}$) using an Nvidia RTX A2000 Laptop GPU.

| | Train | | | | | |
|---|---|---|---|---|---|---|
| $n$ | $A_n$ | $D_n$ | $E_n$ | $\tilde{A}_{n-1}$ | $\tilde{D}_{n-1}$ | $\tilde{E}_{n-1}$ |
| 7 | 150 | 246 | 416 | 340 | 146 | 132 |
| 8 | 442 | 810 | 1,574 | 1,265 | 504 | 1,080 |
| 9 | 1,424 | 2,704 | — | 4,582 | 1,868 | 4,376 |
| 10 | 4,522 | 9,252 | 10,906 | 16,382 | 6,864 | — |
| | Test | | | | | |
| 11 | 14,924 | 32,066 | 24,060 | 63,260 | 25,810 | — |

*Table 2.* Number of quivers of each type and size in train and test sets.

# D. The Mutation Class of $\tilde{D}_{n-1}$ Quivers

Theorem 6.1 groups the mutation class of $\tilde{D}_{n-1}$ quivers into three families: paired types, quivers with one central cycle, and quivers with two central cycles. A complete list of paired types is shown in Figure 13. Here we provide additional details about the subtypes with one and two central cycles.

### D.1. One central cycle

**Types V, Va, Vb.** Type V quivers resemble Type IV quivers of class $D$, but one edge in the central cycle is part of a $B_{\mathrm{IV}}$ block that appears in the Type II quiver of class $D$. In the diagram below, this is the edge $\alpha : a \to b$. Note that no larger subquiver may be attached to $d$ and $d'$.

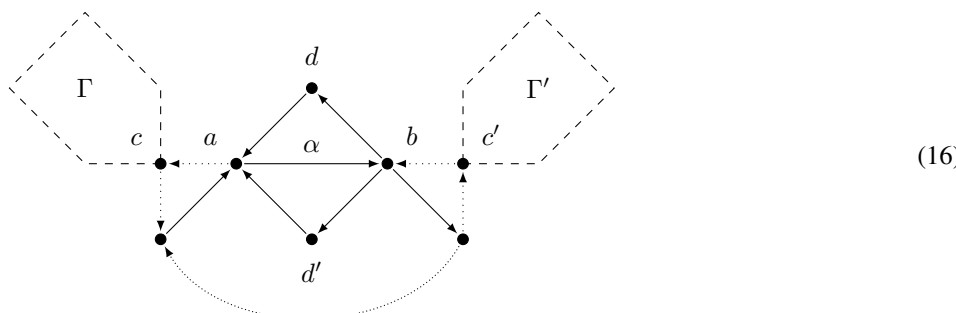

$$(16)$$

Mutating Type V at $d$ produces the subtype **Type Va**, which is similar, but the block $B_{\mathrm{IV}}$ is replaced with an oriented 4-cycle, as shown below.

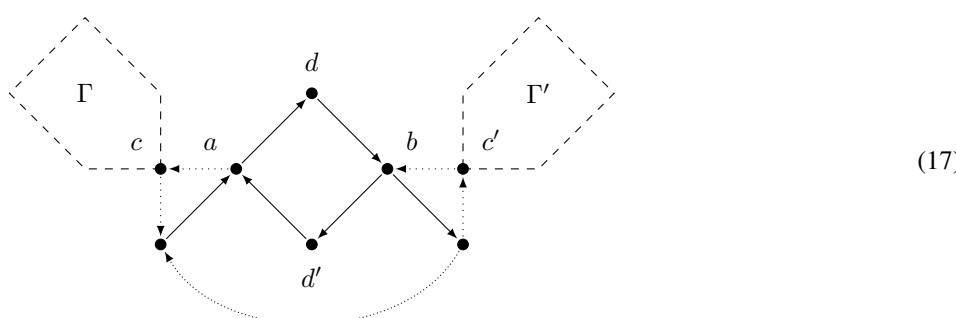

$$(17)$$

Mutating type Va at $d'$ produces the subtype **Type Vb**, which is similar to (16) except the block $B_{\text{IV}}$ is reversed.

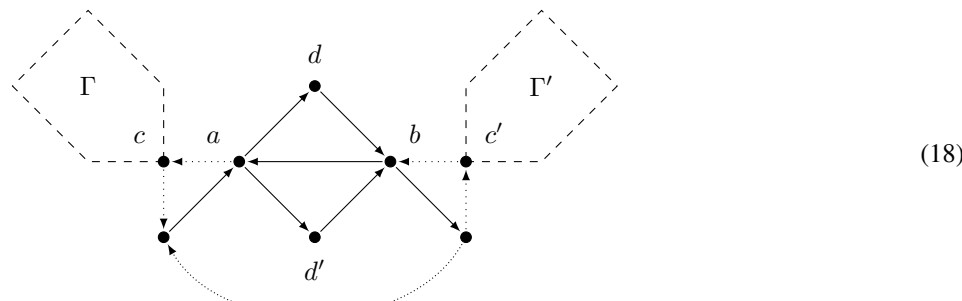

$$(18)$$

Mutating again at $d$ creates a quiver isomorphic to Type Va by swapping $d$ and $d'$, and finally mutating once more at $d'$ results in Type V again. Notice that the choice of $d$ and $d'$ is arbitrary.

**Types V', Va', Vb'.** The rest of this cluster consists of the following types, which we call V', Va', and Vb' as they are related to each other by an analogous sequence of mutations. That is, starting from V', performing the sequence of mutations $\mu_d, \mu_{d'}, \mu_d, \mu_{d'}$ yields Types Va', Vb', Va', V', in that order. Moreover, $\mu_c$ converts each of Type V', Va', Vb' into a corresponding Type V, Va, or Vb quiver, respectively. The Type V' to Type V case is shown in Figure 11.

**Type V'**                          **Type Va'**                          **Type Vb'**

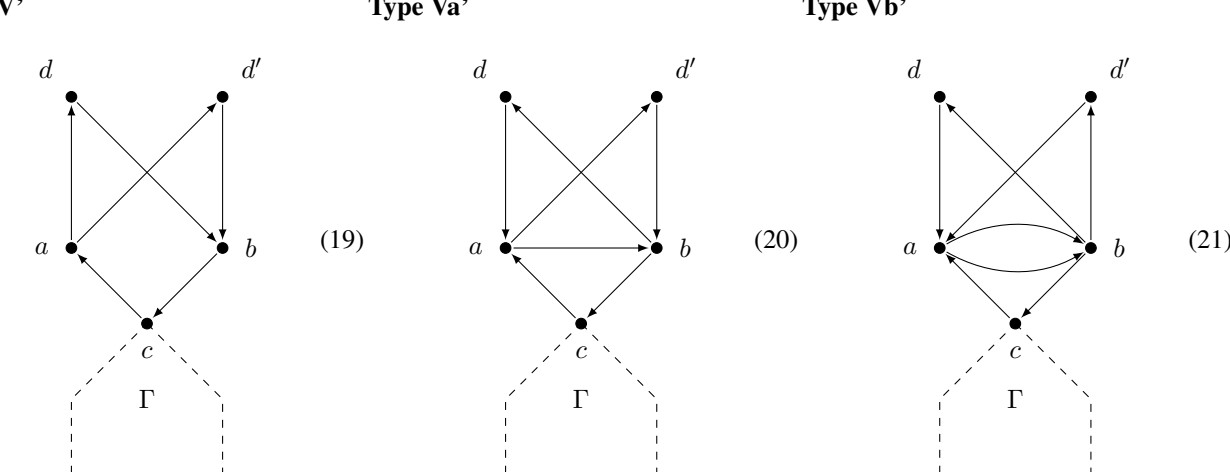

To see that these types are mutation equivalent to (7), it suffices to show that Type V are mutation equivalent to one of the paired types.

**Lemma D.1.** *Type V quivers* (16) *are mutation equivalent to* (7).

*Proof.* We mutate at vertex $a$. There are several cases. Recall from Theorem 5.3 that a *spike* refers to an oriented triangle on the central cycle.

If the central cycle is of length $> 3$, there are two subcases:

(a) If there is a spike at vertex $c$, then the resulting quiver is of Type II-IV, where the vertices $c$ and $a$ are playing the roles of $c$ and $c''$ in Figure 13, respectively, and $b$ plays the role of $c''$.

(b) Otherwise, the resulting quiver is of Type I-IV, where $d$ and $d'$ are the pair of dead ends.

If the central cycle is length 3, say a triangle $a \to b \to v \to a$, then there are four subcases, depending on the presence of spikes on the central cycle:

(a) If the central cycle has no additional spikes, then $\mu_a$ yields a Type I-I quiver.

(b) If $a$ is part of a spike but $b$ is not, then the result is a Type I-II quiver, where $b$ and $v$ are the pair of dead ends on the Type I side and $d, d'$ are the dead ends in the $B_{IV}$ block in the Type II side.

(c) If $a$ is not part of a spike but $b$ is then the result is Type I-III, where $d$ and $d'$ are the pair of dead ends in the Type I side.

(d) If both $a$ and $b$ are parts of spikes, then the result is Type II-III with dead ends $d, d'$ in the $B_{IV}$ block in the Type II side.

$\square$

**Corollary D.2.** *Quivers of Types Va* (17) *and Vb* (18) *are mutation equivalent to* (7).

**Corollary D.3.** *Type V' quivers* (19) *are mutation equivalent to* (7).

**Corollary D.4.** *Quivers of Types Va'* (20) *and Vb'* (21) *are mutation equivalent to* (7).

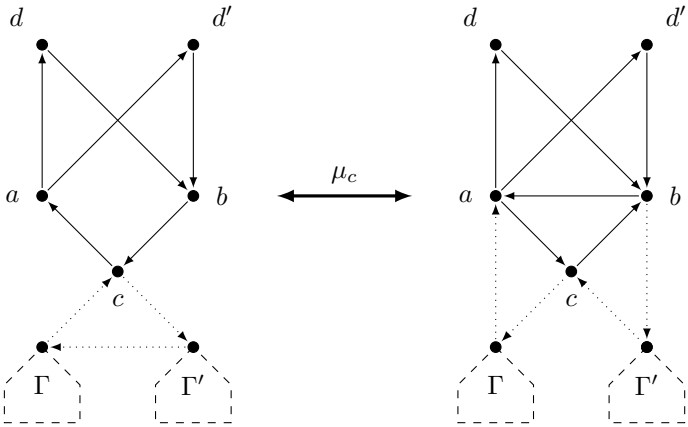

*Figure 11.* Performing $\mu_c$ to convert between a quiver of Type V' (left) and Type V (right). Note that in the Type V quiver, the central cycle is $a \to c \to b \to a$, so the vertex $c$ is not a connecting vertex as it is in 16.

This completes the description of new types with one central cycle.

### D.2. Two central cycles

The other family, with quivers with two central cycles, consists of the following:

**Type VI.** Quivers of Type VI consist of two Type IV quivers which share one vertex $c$ among both central cycles, and are further joined by two edges that create oriented triangles for which $c$ is a vertex. These oriented triangles can be seen as shared spikes among both central cycles. In (22), we color one central cycle blue and one red for clarity. The central cycles

may be any length $\geq 3$.

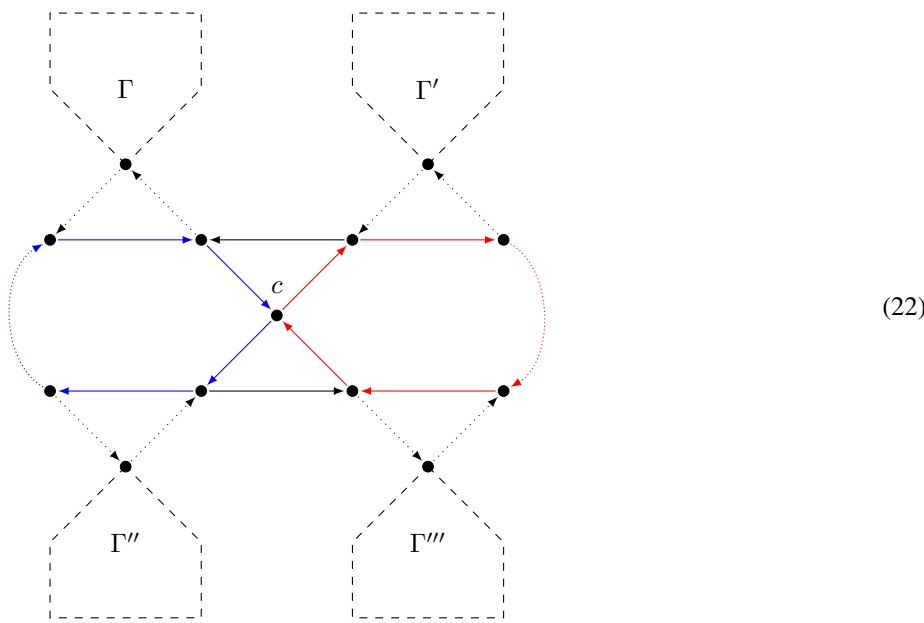

(22)

**Type VI'.** In Type VI, the shared connecting vertex $c$ is not allowed to be a connecting vertex for a larger subquiver of Type $A$ in general. However, there is one exception to this when both central cycles are triangles and have no additional spikes. The result is a block $B_V$ whose outlet is a connecting vertex for a type $A$ subquiver $\Gamma$. We refer to this as Type VI'. Notice that if the quiver has only five vertices, then $\Gamma$ is only one vertex, in which case there is no difference between Types VI and VI'.

(23)

Again, to show that these are mutation equivalent to (7), it suffices to reduce to the paired types.

**Lemma D.5.** *Type VI quivers* (22) *are mutation equivalent to* (7).

*Proof.* If both central cycles are of length $> 3$, then mutating at vertex $c$ results in a quiver Type IV-IV. If a central cycle is of length 3, then $\mu_c$ turns that central cycle into a subquiver of Type I or Type III, depending on whether or not that central cycle does not have or does have a third spike, respectively. □

For the case of Type VI' quivers, the following lemma from (Vatne, 2010) will be useful:

**Lemma D.6** (Vatne 2010). *Let $\Gamma \in \mathcal{M}_n^A$, $n \geq 2$, and let $c$ be a connecting vertex for $\Gamma$. Then there exists a sequence of mutations on $\Gamma$ such that:*

(i) *$\mu_c$ does not appear in the sequence (that is, we do not mutate at $c$);*

(ii) *The resulting quiver is isomorphic to* (1) *in Section 5.1;*

(iii) *Under this isomorphism, $c$ is mapped to 1.*

**Lemma D.7.** *Type VI' quivers* (23) *are mutation equivalent to* (7).

*Proof.* By Lemma D.6, we may mutate so that $c$ has in-degree 0 and out-degree 1 in $\Gamma$. Then mutating at vertex $c$ results in a quiver of Type I-II (cf. Figure 12). □

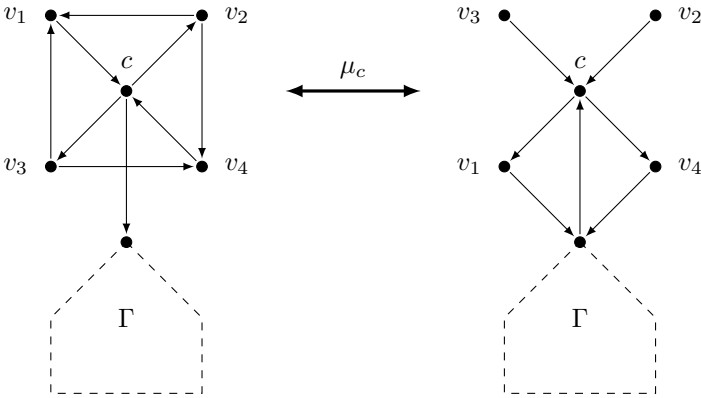

*Figure 12.* Performing $\mu_c$ on a quiver of Type VI' (left).

*Remark* D.8. Our characterization was achieved independently, without knowledge of that of Henrich (2011). Although we will not discuss the overlap in detail, we give a brief description of how our characterization corresponds to (Henrich, 2011), using the notation found therein:

- $D_{\star,\star'}$ in Henrich (2011) corresponds to our paired types, except when $(\star, \star') = ((\bigcirc, n), (\bigcirc, m))$ and $Q$ is in a so-called "merged stage". We call this type VI.

- $D_{(\bigcirc,n)\wedge\square}$, $D_{(\bigcirc,n)\wedge\boxslash}$, $D_{(\bigcirc,n)\wedge\overleftrightarrow{\boxslash}}$ are our types Va, V, Vb, respectively.

- $D_{\square\wedge\square}$ and $D_{\boxslash\wedge\boxslash}$ are our types V' and Vb'. Our type Va' falls into the definition of $D_{(\bigcirc,3)\wedge\square}$.

- $D_{\boxtimes}$ is our type VI'.

# E. Proof of Theorem 6.1

*Proof.* From the preceding lemmas we know that these types are mutation equivalent to the quiver in (7), so we need only prove that these types are exhaustive by showing that $\mathcal{M}_{n-1}^{\tilde{D}}$ is closed under quiver mutation.

We will begin with the paired types. Suppose that $Q \in \mathcal{M}_{n-1}^{\tilde{D}}$ is the union of two quivers $Q_1, Q_2 \in \mathcal{M}^D$ whose intersection $\Gamma_c$ is in $\mathcal{M}^A$. If $\Gamma_c \in \mathcal{M}_k^A$ for $k > 1$, then any mutation can affect the type of at most one of $Q_1$ or $Q_2$ and hence by Theorem 5.3 results in a quiver of (possibly different) paired type. Thus in what follows we assume that $\Gamma_c$ is a single vertex $c$. Moreover, we need only consider mutating at $c$, since a mutation anywhere else can only convert $Q$ from one paired type to another.

In the casework below, when a type V quiver has a central cycle of length 3 and only 7 edges, we will refer to it as *minimal type V*. Similarly, a minimal type VI quiver is a type VI quiver where both central cycles are length 3.

**Type I-I.** Because we assume $\Gamma_c$ is a single vertex $c$, the underlying graph of this quiver is the star graph on 5 vertices, rooted at $c$. Mutating at $c$ depends on the number of arrows to and from $c$. If $c$ has indegree 4 or outdegree 4, then $\mu_c$ simply reverses every arrow. If $c$ has indegree 3 or outdegree 3, then $\mu_c$ produces a minimal Type V quiver. If $c$ has indegree 2 and outdegree 2, then $\mu_c$ produces Type VI (or VI', since they are the same when there are only 5 vertices).

**Type I-II.** Let $c'$ denote the connecting vertex opposite $c$ in the block $B_{IV}$ in the Type II subquiver, and $a, b$ denote the endpoints of the Type I arrows. Then if $Q$ contains the paths $a \to c \to c'$ and $b \to c \to c'$ or $c' \to c \to a$ and $c' \to c \to b$, then $\mu_c$ yields another Type I-II quiver. If $Q$ has $c \to c'$ with $c \to a$ and $c \to b$, or if $Q$ has $c' \to c$ with $a \to c$ and $b \to c$ the result is Type VI'. If $Q$ has $a \to c$ and $c \to b$ (or vice versa) then the result is Type V (regardless of the orientation of the $B_{IV}$ block.

**Type I-III.** If the Type I arrows are of the same orientation with respect to $c$, then $\mu_c$ results in a quiver of Type V. Otherwise, the result is Type VI.

**Type I-IV.** If the Type I arrows are of the same orientation with respect to $c$, then $\mu_c$ results in a quiver of Type V. Otherwise, the result is Type VI.

**Type II-II.** Let $c'$, $c''$ denote the connecting vertices opposite $c$ in the $B_{\mathrm{IV}}$ blocks in the Type II subquivers $Q_1$ and $Q_2$, respectively. Then if the arrows are oriented $c' \to c \to c''$ (or the reverse) then $\mu_c$ yields a quiver of Type VI' where the connecting vertex $c$ is glues the $B_{\mathrm{V}}$ block to an oriented triangle. Otherwise $\mu_c$ yields another Type II-II quiver.

**Type II-III.** Mutating at $c$ yields a Type V quiver (regardless of the $c - c'$ orientation).

**Type II-IV.** Mutating at $c$ yields a Type V quiver (regardless of the $c - c'$ orientation).

**Type III-III.** Mutating at $c$ yields a Type VI quiver with two central cycles of length 3.

**Type III-IV.** Mutating at $c$ yields a Type VI quiver.

**Type IV-IV.** Mutating at $c$ yields a Type VI quiver.

Having finished the paired types, we turn our attention to our newly identified types.

**Type V.** In the proof of Lemma D.1, we showed that mutating at $a$ stays in $\mathcal{M}_{n-1}^{\tilde{D}}$, producing a quiver of paired type in all cases. In Appendix D.1, we showed that mutating at $d$ (and $d'$ by symmetry) produces a Type Va quiver. If we mutate at $b$, we bifurcate into the same cases as when mutating at $a$ in Lemma D.1, and in fact obtain the same types. Now, if the central cycle is of length > 3, then we are done, as mutating anywhere along the central cycle will simply shrink the central cycle by 1, leaving us with another quiver of Type V. However, suppose the central cycle is of length 3, given by $a \xrightarrow{\alpha} b \to v \to a$. Then we must consider $\mu_v$, which yields Type V'. (In this manner, Type V' can be seen as the result of shrinking the central cycle to length 2, which we then remove because digons are prohibited.) Here the subquiver $\Gamma$ is a single vertex if there are no additional spikes, a single directed edge if there is one spike, and an oriented triangle if there are two spikes.

**Type Va.** We know from Appendix D.1 that $\mu_d$ and $\mu_{d'}$ yield Types V and Vb, respectively. Continuing, $\mu_a$ and $\mu_b$ both yield Type VI. If the central cycle (sans $\alpha$) is length > 3, then we are done. Otherwise, we consider the case where we have a vertex $v$ with $b \to v \to a$ and compute $\mu_v$, which we see yields Type Va'.

**Type Vb.** From Appendix D.1 we see that $\mu_d$ and $\mu'_d$ yield Type Va. Mutating at $a$ or $b$ yields Type Vb again, simply moving the reversed arrow around the central cycle with the associated $d, d'$. Finally, we are done unless the central cycle is an unoriented triangle $a \leftarrow b \to v \to a$, in which case we must consider $\mu_v$, which yields Type Vb'.

**Type V'.** We know the mutations $\mu_d$ and $\mu_{d'}$ yield Type Va'. The mutations $\mu_a$ and $\mu_b$ yield Type VI'. Finally, we know that mutating at $c$ yields Type V from Corollary D.3 (see Figure 11).

**Type Va'.** As we have seen, $\mu_d$ yields Type V' and $\mu_{d'}$ yields Type Vb'. The mutations $\mu_a$ and $\mu_b$ both result in Type Va' by cyclically permuting $(a, b, d)$ forwards and backwards, respectively. Finally, $\mu_c$ yields Type Va.

**Type Vb'.** The mutations $\mu_d$ and $\mu_{d'}$ yield Type Va'. Mutating at $a$ or $b$ produces an automorphism which swaps $a$ and $d$ with $b$ and $d'$, respectively, so $\mu_a$ and $\mu_b$ yield Type Vb' again. Finally, mutating at $c$ yields Type Vb.

**Type VI.** From Lemma D.5 we know that $\mu_c$ yields a paired type. Call the two central cycles $C_1$ and $C_2$. If we mutate at any vertex which is not adjacent to $c$, the result is still Type VI, as the mutation affects the relevant cycle $C_1$ or $C_2$ as a Type IV subquiver, and cannot break $C_1$ or $C_2$. Suppose then that we mutate at a vertex $v$ which is adjacent to $c$ and suppose, without loss of generality, that $v \in C_1$. Then $\mu_v$ simply moves $v$ from $C_1$ to $C_2$, resulting in Type VI, unless $C_1$ is a triangle, in which case the result is Type Va.

**Type VI'.** Since $c$ is a connecting vertex in $\Gamma$, it has degree at most 2. If $c$ has degree 1 in $\Gamma$, $\mu_c$ yields Type I-II, and if $c$ has degree 2 in $\Gamma$, $\mu_c$ yields Type II-II. Any other mutation in $\Gamma$ cannot change the type, so it remains only to check the vertices adjacent to $c$. Mutating at any results in Type V'.

Thus we have shown that $\mathcal{M}_{n-1}^{\tilde{D}}$ is closed under quiver mutation. This completes the proof. $\qquad\square$

# F. Additional figures

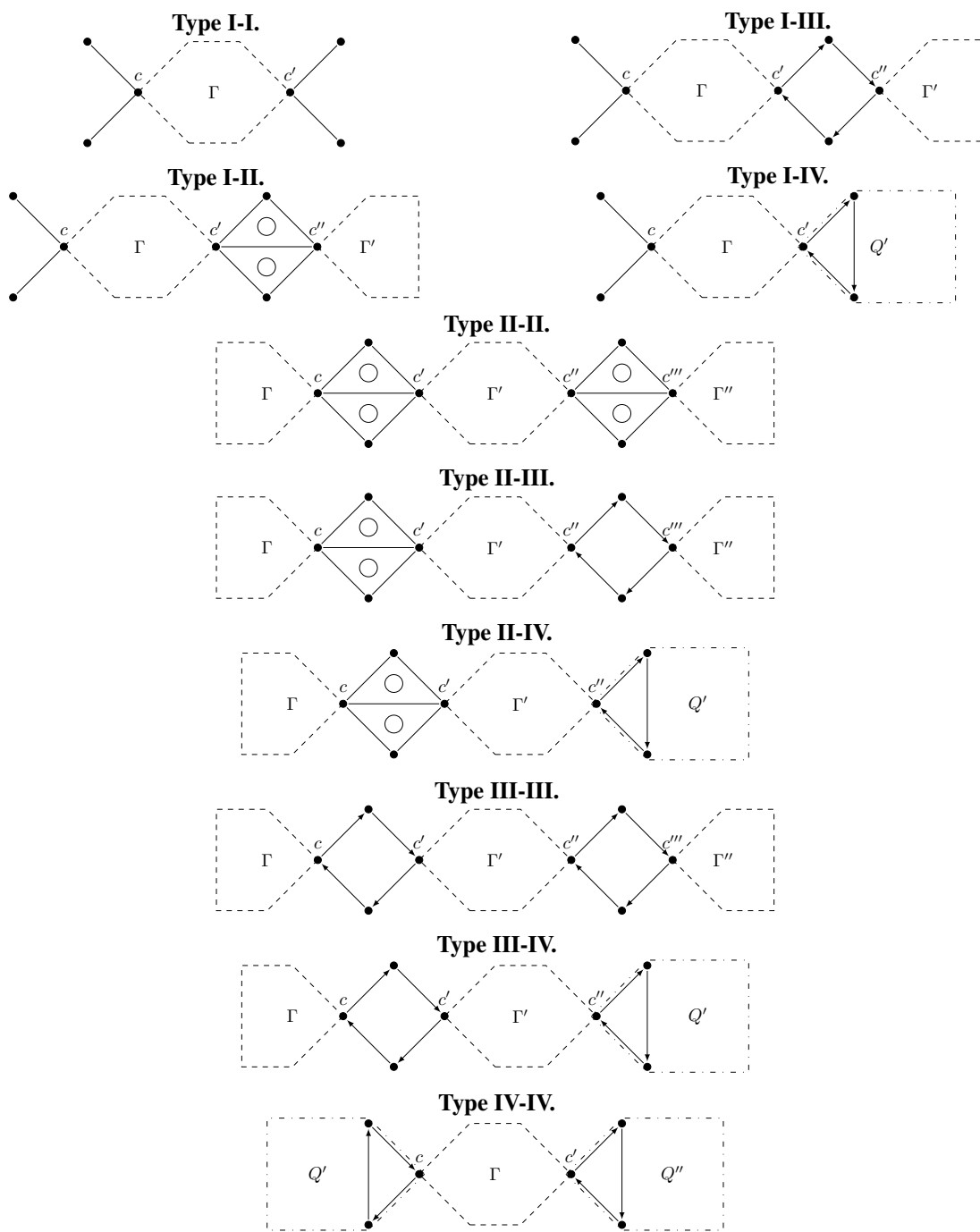

*Figure 13.* All paired types. Unoriented edges may have any orientation. Circles indicate oriented cycles. Here $\Gamma$, $\Gamma'$, $\Gamma''$ are subquivers of Type $A$, and $Q'$ is a subquiver of Type $D$-IV for which $c, c'$, or $c''$ is part of a spike. Notice that we may have $\Gamma \in \mathcal{M}_1^A$ with $c = c'$, but $Q'$ and $Q''$ must contain at least two edges in addition to the ones shown.

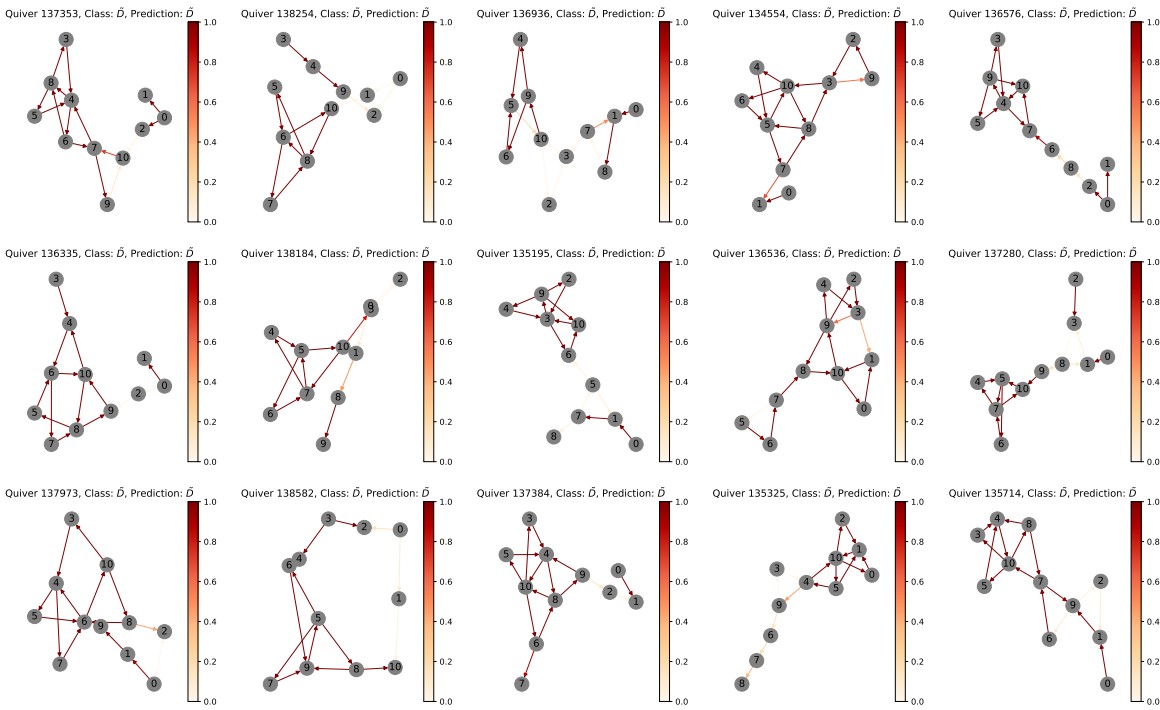

*Figure 14.* Randomly selected quivers from the orange (left) cluster in Figure 8, consisting of quivers of Types V, Va, Vb, V', Va', Vb'.

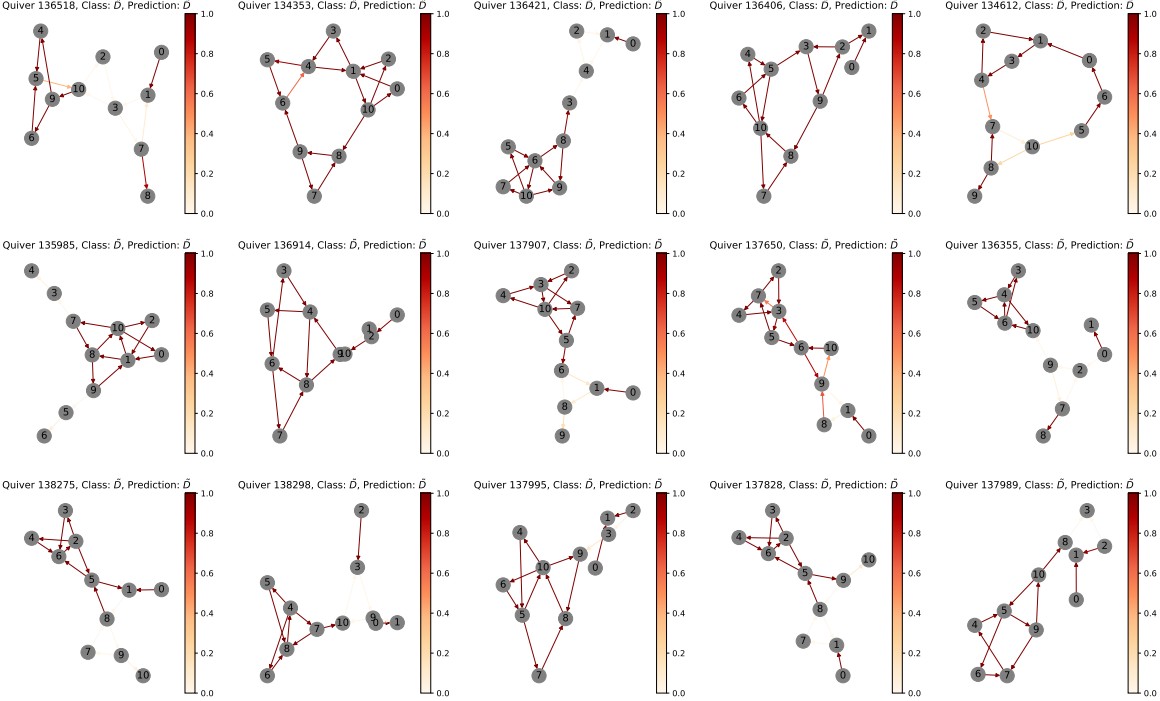

*Figure 15.* Randomly selected quivers from the blue (right) cluster in Figure 8, which consists of quivers of Types VI and VI'.

