# OpenReview forum: "Machines and Mathematical Mutations: Using GNNs to Characterize Quiver Mutation Classes"
_ICML.cc/2025/Conference — ICML 2025 poster_

### Official Review · Reviewer_E1Ro · 2025-03-04

**Overall Recommendation:** 3

**Summary:**

In this paper, the authors study the problem of quiver mutation using the GNNs and GNN explanation tool. The author identified that the GNNs trained with naive classification tasks on predicting quiver mutation type are able to learn causal information related to quiver mutation. In particular, it can identify some existing theorems without any prior knowledge.

## Update after rebuttal
I would like to thank the authors for the detailed response to my questions and concerns. Most of my concerns are appropriately addressed by the rebuttal. I would like to keep my original score.

**Claims And Evidence:**

The overall procedure makes sense to me, and using the algorithm alignment ability of GNNs to solve mathematic problems is interesting.

**Essential References Not Discussed:**

No.

**Experimental Designs Or Analyses:**

See above.

**Methods And Evaluation Criteria:**

1. I am not really familiar with the quiver mutation. But I am just wondering if there is any additional concern that the dataset only contains graphs of size 6-10 in training and 11 for testing. Is it possible to evaluate on a graph with a size much larger than training (like 20 or 30), and will the conclusions still hold?
2. The analysis only focuses on the type $D$ and $\widetilde{D}$, Is it possible to also extend the analysis to other types and discover other patterns?
3. The paper only examines several particular theorems. I am wondering how we can generalize results to other problems, especially for problems where we do not know whether GNN is useful. As the finally goal is that we can use ML models to solve mathematical problem or identify theorem we are unknown, I think how to systematically classify a certain problem to corresponding ML class that is algorithmic alienable, how to train the ML model and how to leverage explanation tool to discover new knowledge would be more interesting and impactful.

**Other Comments Or Suggestions:**

See above

**Other Strengths And Weaknesses:**

See above

**Questions For Authors:**

See above

**Relation To Broader Scientific Literature:**

The method and pipeline used in the paper may be applicable to other mathematic problems.

**Theoretical Claims:**

As I am not familiar with Quiver mutation, I cannot assess the most theoretical claims made in the paper.

---

> ### Author Rebuttal · Authors · 2025-03-31
>
> Thank you for your feedback and insightful questions. We're glad to hear you found the topic interesting, and we hope to answer your questions below:
>
> > Is it possible to evaluate on a graph with a size much larger than training (like 20 or 30), and will the conclusions still hold?
>
> The properties of each mutation equivalence class remain valid *regardless of the size*, so from the domain perspective the exact size of the graphs in the test set is not particularly important.
>
> From the machine learning perspective, we can certainly evaluate the model on larger graphs. Below we test the same model checkpoint on quivers of up to 20 nodes. Because the number of distinct quivers grows very quickly with size, we can only check on a subsample (we use a mutation depth of 6 and, if necessary, randomly sample 100,000 quivers to avoid out-of-memory errors).
>
> | Nodes | Accuracy |
> |:-----:|:--------:|
> | 12    | 99.6%    |
> | 13    | 98.7%    |
> | 14    | 97.7%    |
> | 15    | 95.5%    |
> | 16    | 94.3%    |
> | 17    | 92.0%    |
> | 18    | 91.1%    |
> | 19    | 89.4%    |
> | 20    | 89.1%    |
>
> The model continues to perform well though performance does begin to degrade. We believe this is because the GNN has a fixed depth of 4, and hence may struggle to capture certain larger non-local substructures that appear in quivers with more nodes (e.g. long cycles). We discuss the theoretical implications of using a message-passing GNN for large quivers in the Response to Reviewer ivH2.
>
> > The analysis only focuses on the type $D$ and $\widetilde{D}$, Is it possible to also extend the analysis to other types and discover other patterns?
>
> It should be possible to extract the characterizations of other finite mutation classes in a similar manner, recovering other results from Henrich [1]. Certain mutation-infinite classes may also admit characterization by certain patterns, though it is not known in general which other mutation classes, if any, admit such a characterization. If such a characterization does exist, it may be possible to extract these from a machine learning model as well.
> However, we note that mutation equivalence is in general a very hard problem [2], so discovering patterns for an arbitrary mutation class is likely also difficult in general.
>
> > I think how to systematically classify a certain problem to corresponding ML class that is algorithmic alienable, how to train the ML model and how to leverage explanation tool to discover new knowledge would be more interesting and impactful.
>
> We agree, and we think this is an interesting direction for future work. GNNs are a natural choice for many problems, but there is emerging evidence for reasoning capabilities in transformers, for example. A more general systematic pipeline, however, will require a more mature suite of interpretability tools and a deeper understanding of algorithmic alignment in different architectures.
>
> [1] Henrich, Mutation-classes of diagrams via infinite graphs, Math Nachr., 2011
> [2] Soukup, Complexity of quiver mutation equivalence, preprint, 2023

---

### Official Review · Reviewer_WJi6 · 2025-03-12

**Overall Recommendation:** 2

**Summary:**

This work uses GNNs to solve the quiver-mutation-equivalence problem: whether one quiver can be transformed into the other through a sequence of mutations. With explainability techniques, they discovers criteria for quiver of type $\tilde D$. Moreover, GNN need not to be trained.

**Claims And Evidence:**

Yes.

**Essential References Not Discussed:**

The GNN and Explainablity methods used in the paper are discussed in method or Appendix section, but not introduced in related work section.

**Experimental Designs Or Analyses:**

Yes.  The training set contains quiver with nodes 6-10, while test set containing quiver with 11 nodes only, which avoids the risk of data leakage.

**Methods And Evaluation Criteria:**

Yes. The proposed GNN is a simple adaptation of GIN to directed multigraph settings.

**Other Comments Or Suggestions:**

In line 157, please refer to figure 3 for a more clear illustration of quiver type A,D,E ...

**Other Strengths And Weaknesses:**

This work is more similar to a experiment report rather than a paper, as no novel algorithm or tasks are proposed, and the result is used to verified existing results rather than new results.

**Questions For Authors:**

Equivalence relation is more similar to a contrastive learning task rather than classification task. Have you even tried formulate the task as predicting whether two graph are equivalent rather than predict the type of one graph?

**Relation To Broader Scientific Literature:**

As the author claims, it is related to the quiver mutation problem.

**Theoretical Claims:**

No. All proofs are on the property of quiver mutations, which are primary the background problem setting rather than the method. I do not have the domain knowledge and skip it.

---

> ### Author Rebuttal · Authors · 2025-03-31
>
> We appreciate your feedback and your suggestions regarding presentation. Below we hope to address your concerns and questions.
>
> > This work is more similar to a experiment report rather than a paper, as no novel algorithm or tasks are proposed, and the result is used to verified existing results rather than new results.
>
> Although our work is an application-driven case study rather than a specific novel method, we believe the application of existing machine learning techniques to the quiver mutation equivalence problem is novel and has potential impact in AI for mathematics. While our mutation equivalence results were known, we reiterate that *we discovered Theorem 5.1 independently of prior work*.
>
> As we have also discussed in or response to Reviewer ivH2, the use of ML to accelarate scientific discovery is of great interest to the machine learning community. Indeed, it is often pointed to as one of the primary positive impacts that modern machine learning can provide. Understanding how ML tools should be used to enable scientific discovery must be a conversation that involves both domain and ML experts. As such, we believe that works that illuminate this process are very much in line with the role of ICML in the ML community.
>
> > In line 157, please refer to figure 3 for a more clear illustration of quiver type A,D,E ...
>
> Thank you for the suggestion. We will move the reference to Figure 3 from line 142 to line 157.
>
> > Equivalence relation is more similar to a contrastive learning task rather than classification task. Have you even tried formulate the task as predicting whether two graph are equivalent rather than predict the type of one graph?
>
> The contrastive learning task is an interesting question in its own right. However, our goal of extracting structural characterization results from the model motivated our choice to formulate the task as classification. From an explainability perspective, we are interested in the question "Why does the GNN classify this as Type $D$?" rather than the question "Why does the GNN predict these two quivers are mutation equivalent?" Thus we chose to formulate the machine learning task as a classification problem rather than a contrastive problem.

---

### Official Review · Reviewer_KQ27 · 2025-03-14

**Overall Recommendation:** 4

**Summary:**

This paper shows that a GNN learns the same substructure to classify the quiver mutation class of a quiver as proposed by a classification theorem in quiver theory in mathematics. The authors train a GNN on quivers of different types $A,D,E,\tilde{A},\tilde{D},\tilde{E}$ and use PGExplainer on this trained model to detect the substructure that the model is focussing on, in order to classify the mutation class of quivers. Further, in section 5 they conjecture a theorem which they prove about the mutation class of quivers of type $\tilde{D}$.

**Claims And Evidence:**

The claims made in the paper are supported by sufficient evidence.

**Essential References Not Discussed:**

I don't think so.

**Experimental Designs Or Analyses:**

Why do you choose to train on A,D,E,$\tilde{A}$,$\tilde{D}$,$\tilde{E}$ when the test set is not going to contain any samples from $\tilde{E}$?

Moreover, what is the reason behind training on quivers with lesser nodes and testing on quivers of a fixed node size which is larger than what the model has been trained on?

**Methods And Evaluation Criteria:**

A holistic viewpoint of the methods seems to make sense for the problem at hand. However, some finer details such as choice of design of experiments do not seem to be motivated enough, as pointed out in the later section of the review.

**Other Comments Or Suggestions:**

You might want to add the fact that the message passing is being done over graphs with constant node features to highlight the fact that the GNN model is truly learning only the structure with little to no help from the node features.

**Other Strengths And Weaknesses:**

Strengths:

Paper is well-written and well-organized.

Weaknesses:

As pointed out earlier, the choice of the design of experiments does not seem to be motivated enough.

**Questions For Authors:**

Did you try generating the graphs with different features? E.g. Node degrees or similar features?

**Relation To Broader Scientific Literature:**

I think this work shows that machine-guided research is an avenue that needs to be explored more by the AI for science community. Though the specific task that the authors choose to show this might seem niche, the idea that machine recognizes the same substructures in a problem that a classification theorem gives is really powerful.

**Theoretical Claims:**

Yes, I checked the proof of Theorem 5.1, and as a consequence Lemmas D.1, D.5-D.7 and Corollaries D.2 - D.4 in Appendix D.

---

> ### Author Rebuttal · Authors · 2025-03-31
>
> We appreciate your insightful review. We are encouraged you think our work demonstrates the promise of machine-guided research, and we are glad you found the paper well-organized. Since multiple reviewers had questions about the experimental design, we have also expanded the discussion in the paper to address the most common questions and emphasize that our experimental design was driven by our ultimate goal of generating domain-related insight from the model.
>
> We aim to address your questions more specifically below:
>
> > Why do you choose to train on A,D,E, $\widetilde{A}$, $\widetilde{D}$, $\widetilde{E}$, when the test set is not going to contain any samples from $\widetilde{E}$?
>
> We train the model on entire mutation classes of smaller sizes to ensure the model sees a comprehensive view of each mutation class during training. Our inclusion of types $E$ and $\widetilde{E}$ in the train set primarily serves to increase the difficulty of the training task, since ultimately our analysis focuses on types $D$ and $\widetilde{D}$. (From the domain perspective, there are only finitely many $\widetilde{E}$ of any size, so general classification can be achieved through exhaustive computation, and the class $E$ is no longer mutation-finite for larger sizes, making a combinatorial characterization substantially more difficult.)
>
> > Moreover, what is the reason behind training on quivers with lesser nodes and testing on quivers of a fixed node size which is larger than what the model has been trained on?
>
> The difference in sizes from train to test is in part a consequence of our desire to train on entire mutation equivalence classes, as we mentioned previously. In addition, from the application perspective any classification result for a mutation class should generalize across sizes. In this manner the difference in graph sizes follows prior work studying size generalization of GNNs (e.g. [1]).
>
> > You might want to add the fact that the message passing is being done over graphs with constant node features to highlight the fact that the GNN model is truly learning only the structure with little to no help from the node features.
>
> Thank you, this is an insightful suggestion! We have added this to the paper.
>
> > Did you try generating the graphs with different features? E.g. Node degrees or similar features?
>
> We did not generate graphs with node features. As you point out, we used constant node features so that the GNN learned entirely from the graph structure. In particular, our ultimate goal was to provide structural characterizations of mutation classes, so we wished to ensure that the GNN was truly relying on the graph structure to perform classification.
>
> [1] Yehudai et al., From Local Structures to Size Generalization in Graph Neural Networks, ICML 2021

---

> > ### Comment · Reviewer_KQ27 · 2025-04-07
> >
> > I thank the authors for their response. I would like to maintain my score.

---

### Official Review · Reviewer_ivH2 · 2025-03-14

**Overall Recommendation:** 3

**Summary:**

The paper uses GNNs to learn quiver mutation equivalence. The results show that GNNs can not only classify these quiver types, but can also characterize particular mutation classes through their latent representations.

**Claims And Evidence:**

Most claims are supported by evidence including theoretical results, experiments or case analysis.

**Essential References Not Discussed:**

N/A

**Experimental Designs Or Analyses:**

The paper only uses one type of classical GNN architecture, without investigating more expressive and more state-of-the-art GNN variants. Simultaneously, the explanation method only involves PGExplainer, while there are also many other GNN explainers. Moreover, the small model size and inadequate training does not match usual deep learning schemes, which make the results less convincing.

**Methods And Evaluation Criteria:**

As admitted in the paper, the gap between train and test set (including the distribution of the number of nodes, and the absence of $\tilde E$ in the test set) does not follow common machine learning settings. All the experiments are carried on one dataset, while more comprehensive will be more convincing.

**Other Comments Or Suggestions:**

This paper is essentially a naïve application of simple machine learning methods to one specific problem, without strong theoretical guarantees and large scale experimental verifications. My opinion is that it is not yet qualified for a top-tier AI/machine learning conference like ICML, and would suggest submitting the paper to some other conferences/journals for data analysis or applied mathematics.

**Other Strengths And Weaknesses:**

The paper is an interesting application of GNNs to the mutation equivalence problem. However, the machine learning techniques in this paper are elementary, and the analytical results lack statistical guarantee over larger scale.

**Questions For Authors:**

* Can you provide analysis on theoretical expressive power of directed GNNs, or at least for distinguishing quiver mutation classes?

* Can you elaborate on more large-scale/real-world  datasets and more state-of-the-art models?

**Relation To Broader Scientific Literature:**

The paper is related to literature in mathematics, topology and graph neural networks.

**Theoretical Claims:**

I check the correctness and did not find major flaws. However, a great proportion of definitions (those in Section 3) and theoretical results (e.g., Theorem 4.3) are known results or trivial extensions, which bothers reading and may downgrade the contribution of the paper. While there are tons of theoretical work on the expressivity of graph neural networks, I strongly recommend the authors look deeper into theoretical expressive power of (directed) GNNs to provide theoretical guarantees for GNNs to characterize quiver mutation classes.

---

> ### Author Rebuttal · Authors · 2025-03-31
>
> Thank you for your thoughtful review, and we are glad you found the application to quiver mutation interesting. We hope to address your comments below:
>
> > As admitted in the paper, the gap between train and test set (including the distribution of the number of nodes, and the absence of $\widetilde{E}$ in the test set) does not follow common machine learning settings. All the experiments are carried on one dataset, while more comprehensive will be more convincing.
>
> Regarding the experimental methods, we recognize that the difference in the train and test sets does not follow common machine learning practice. Rather, *our experimental design is motivated by our ultimate goal of recovering classification theorems from the model*. As we discuss in our Response to Reviewer KQ27, the presence of $\widetilde{E}$ in the training set serves to modulate the difficulty of the problem to ensure that the model learns discriminative features for the other classes, particularly $D$ and $\widetilde{D}$, on which we focus our explainability study. Similarly, the difference in the distribution in the number of nodes between training and testing ensures that the GNN learns structural traits which are size-generalizable, as well as allowing us to provide the GNN with the full mutation classes for smaller sizes without contaminating the test data.
>
> > The paper only uses one type of classical GNN architecture, without investigating more expressive and more state-of-the-art GNN variants. Simultaneously, the explanation method only involves PGExplainer, while there are also many other GNN explainers.
>
> Our intention was not to investigate which GNN architecture would be the most effective for classifying quiver mutation classes, nor was it to compare between different explainability methods. Rather, *we provide a case study that illustrates how a domain expert using off-the-shelf components can guide their own research with machine learning*. This application-driven approach differs from works whose goal is to introduce novel architectures or algorithms to be adopted by others. In such settings we agree that it is important to show that the architecture will perform well beyond the initial dataset it was trained for. In our work, however, once the model offers sufficient insight to conjecture and prove a statement about the quiver mutation classes of interest, its performance on additional data ceases to be important.
>
> Ultimately, while we admit the individual methods are not novel in and of themselves, we believe our work presents a novel application of existing methods to an interesting problem in combinatorics. More broadly, we believe the central idea that machine learning can guide domain research is absolutely of interest to the ICML audience.
>
> > Can you provide analysis on theoretical expressive power of directed GNNs, or at least for distinguishing quiver mutation classes?
>
> The question of GNN expressive power (equivalently, the WL test) and quiver mutation is interesting, and we have expanded this discussion in the paper. Directed GNNs are known to be strictly more expressive than undirected 1-WL [1], and we thank you for pointing out this oversight in our discussion and references. In the context of quiver mutation, there are some relevant local substructures which an undirected GNN clearly cannot distinguish. For example, a directed triangle should be of Type $A_3$ while an undirected triangle is of Type $\tilde{A}_2$. In general, the directed WL test is sufficient to recognize type $A$ quivers, as well as certain subtypes of $D$ and $\widetilde{D}$ quivers (namely Types I, II and the corresponding paired types). However, one of the WL test's limitations is its inability to count larger cycles, such as those that appear in Type $D$-IV and $\widetilde{D}$-VI.
>
> > Can you elaborate on more large-scale/real-world datasets and more state-of-the-art models?
>
> GNNs' ability to generalize across sizes and their inductive bias towards local structures also motivates our choice of architecture. While e.g. graph transformers also provide promising reasoning capabilities on graph tasks, message-passing remains the state of the art for recognizing many of the local substructures which are relevant to this problem [2]. However, as we note above, message-passing fails to recognize larger-scale structures in graphs. Our response to Reviewer E1Ro contains an empirical investigation on larger graphs (up to $n = 20$ nodes), revealing that our GNN generalizes well but not perfectly on larger graphs.
>
> [1] Beddar-Wiesing et al., On the Extension of the Weisfeiler-Lehman Hierarchy by WL Tests for Arbitrary Graphs, MLG@ECMLPKDD 2022
>
> [2] Sanford et al., Understanding Transformer Reasoning Capabilities via Graph Algorithms, NeurIPS 2024

---

> > ### Comment · Reviewer_ivH2 · 2025-04-07
> >
> > Thank you for your rebuttal. My concerns are basically addressed. I strongly encourage the authors to include these new results, and acknowledge the limitations as above in their revisions. I raised my scores accordingly.

---

### Decision · Program_Chairs · 2025-05-01

**Decision:**

Accept (poster)

**Comment:**

This paper uses GNNs to learn quiver mutation equivalence. The results show that GNNs can not only classify these quiver types, but can also characterize particular mutation classes through their latent representations.

The reviewers agree that the paper meets the standards of an Application-Driven Machine Learning Paper: novel methods, datasets, tasks, and/or metrics. The reviewers state that the paper is  a well-motivated application paper for applying GNNs to a math problem, and that the authors provide many experimental results to demonstrate that well-trained GNNs can identify existing mathematical results automatically. Moreover, during the rebuttal, the authors also provided additional experiments on the generalization results, which are positive support for their existing results. The authors are asked to include these results in their camera-ready paper if accepted.

The reviewers also state two main limiting factors in the paper: 1) the paper only provides evidence for using GNNs to validate existing results, but does not discover any new scientific results, and 2) the potential audience is limited by the fact that the problem under investigation is somewhat esoteric. Still the very fact that GNNs can discover mathematical results is worthy of publication.